# Predictive variational inference:
# Learn the predictively optimal posterior distribution

**Jinlin Lai** [1]  **Antonio Ricardo Linero** [2]  **Yuling Yao** [2]

## Abstract

Vanilla variational inference finds an optimal approximation to the Bayesian posterior distribution, but even the exact Bayesian posterior is often not meaningful under model misspecification. We propose predictive variational inference (PVI): a general inference framework that seeks and samples from an optimal posterior density such that the resulting posterior predictive distribution is as close to the true data generating process as possible, while this closeness is measured by multiple scoring rules. By optimizing the objective, the predictive variational inference is generally not the same as, or even attempting to approximate, the Bayesian posterior, even asymptotically. Rather, we interpret it as an implicit hierarchical expansion. Further, the learned posterior uncertainty detects heterogeneity of parameters among the population, enabling automatic model diagnosis. This framework applies to both likelihood-exact and likelihood-free models. We demonstrate its application in real data examples.

## 1. Introduction

Traditional Bayesian inference has often been used for one of two purposes: estimation of parameters $\theta$ in a model $p(y|\theta)$ or estimation of a *predictive distribution* $p(\tilde{y}|y)$ to give a probabilistic forecast of a future observation $\tilde{y}$. In many settings, such as decision-making, machine learning, or forecasting, the explicit aim is prediction rather than parameter estimation; fortunately, when both the likelihood $p(y|\theta)$ and prior $p^{\mathrm{prior}}(\theta)$ are correctly-specified, traditional Bayesian inference is optimal for both purposes.

There is, however, an increasing awareness of limitations of

Bayes under misspecification (Walker, 2013; Gelman et al., 2020; Masegosa, 2020; Huggins & Miller, 2023; Fong et al., 2023). Assume the observations $y = (y_1, \ldots, y_n)$ are independent and identically distributed (IID), and let $\tilde{y}$ to be the next unseen data. The Bayesian posterior predictive distribution of $\tilde{y}$ is the sampling distribution averaged over the posterior density: $p^{\mathrm{Bayes}}(\tilde{y}|y) = \int_{\Theta} p(\tilde{y}|\theta)p(\theta|y)d\theta$. But is this exact Bayesian prediction "optimal"? Or is this Bayesian prediction at least more "optimal" than some point-estimate induced prediction $p(\tilde{y}|\hat{\theta})$ where $\hat{\theta}$ is for example the MLE? The answers are in general both negative (Clarke & Yao, 2025), accompanied by real-data evidence that sometimes Bayesian methods instead produce worse predictions (Wenzel et al., 2020) or overconfident inference (Yang & Zhu, 2018). The only predictive optimality we may assert is that, *if the likelihood and prior are both correct*, then among all probabilistic forecasts, the Bayesian posterior prediction minimizes the posterior-averaged KL divergence from the sampling distribution (Aitchison, 1975), i.e.,

$$\arg\min_{Q(\cdot)} \int \mathrm{KL}\left(p(\tilde{y}|\theta)||Q(\tilde{y})\right)p(\theta|y)d\theta = p^{\mathrm{Bayes}}(\tilde{y}|y). \quad (1)$$

But this model-being-correct assumption is unlikely to be relevant to any realistic application. Even if the model is correct, this optimality statement typically does not hold for other divergences practitioners may care about.

To make Bayesian inference under potential model misspecification, this paper develops *predictive variational inference* (PVI): a general framework that constructs predictively optimal posterior samples $q(\theta|y)$. Unlike classical VI, we optimize a divergence $D$ between the *posterior predictive distribution* and the data generating process $p_{\mathrm{true}}(\tilde{y})$:

$$\min_{q \in \mathcal{F}} D\left(\int_{\Theta} p(\tilde{y}|\theta)q(\theta)d\theta \,\Big\|\, p_{\mathrm{true}}(\tilde{y})\right). \quad (2)$$

PVI incorporates various scoring rules, resulting in a family of PVI algorithms each having a statistically meaningful divergence in the objective (2). We typically use a flexible variational family $\mathcal{F}$ such as a normalizing flow (Papamakarios et al., 2021). Rather than viewing this as a sub-optimal approximation to the exact Bayes, PVI directly *learns the optimal predictive distribution* directly. We find three advantages: (a) The PVI-learned optimal predictive is generally

[1]College of Information and Computer Sciences, University of Massachusetts Amherst [2]Department of Statistics and Data Sciences, University of Texas, Austin. Correspondence to: Yuling Yao <yyao@austin.utexas.edu>.

*Proceedings of the 43rd International Conference on Machine Learning*, Seoul, South Korea. PMLR 306, 2026. Copyright 2026 by the author(s).

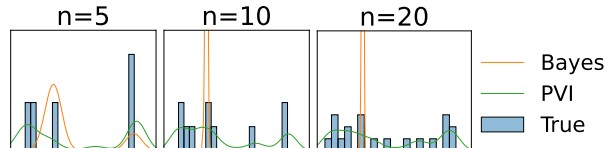

*Figure 1.* Inference for the molecule angle in a heterogeneous population using cryoEM images. For details, see Section 4.3.

different from exact Bayes, even with an infinite amount of data or a flexible enough variational family, while its resulting predictive distribution better fits the observed data. We interpret PVI as an implicit hierarchical expansion. (b) We use PVI as a diagnostic tool. The discrepancy between PVI and exact Bayes is no longer an approximation error; rather, it reveals the *model misspecification*. We use this heuristic to tell which parameter should vary in the population to improve the model. (c) PVI applies to both likelihood-exact and likelihood-free settings. Via a simulation-based divergence $\mathcal{D}$, PVI leads to a computationally efficient tool for simulation-based inference (SBI) tasks. Figure 1 gives a preview of the practical benefits of PVI: We analyze cryogenic electron microscopy (cryoEM) images using a likelihood-free model. The parameter of interest is the bond angle of a molecule, which varies across the population due to molecular heterogeneity. While the exact Bayes posterior quickly collapses to an overconfident point mass, PVI accurately recovers the true population distribution.

### 1.1. Relationship of PVI to VI

Variational inference (VI, Jordan et al., 1999; Blei et al., 2017) is the workhorse of approximate Bayesian posterior inference for large datasets. Given a sequence of observations $y = (y_1, \ldots, y_n)$, $y_i \in \Omega$, and parameter vectors $\theta \in \Theta$, instead of sampling from the exact Bayesian posterior density $p(\theta|y)$, VI usually minimizes the Kullback–Leibler (KL) divergence between an approximate inference $q(\theta)$ and the target posterior density $p(\theta|y)$, among a tractable variational family $q \in \mathcal{F}$, $\min_{q \in \mathcal{F}} \mathrm{KL}\left(q(\theta) \,\big\|\, p(\theta|y)\right)$, which is equivalent to maximizing the evidence lower bound (ELBO), the expected log likelihood plus a prior regularization: $\mathrm{ELBO} = \int_\Theta \sum_{i=1}^n \log p(y_i|\theta) q(\theta) d\theta - \mathrm{KL}(q(\theta)||p^{\mathrm{prior}}(\theta))$.

VI typically underestimates posterior uncertainty, making VI an inferior approximation to the gold standard Markov chain Monte Carlo (MCMC). On the other hand, the uncertainty reflected in the exact Bayes is not always desirable, as Bayesian inference is conditional on the belief model which is nearly always wrong. In particular, when the sample size $n$ is big enough, the Bernstein–von Mises theorem ensures that both VI and the exact posterior concentrate to a point mass around the maximum likelihood estimation (MLE). This vanishing posterior uncertainty compromises the flexi-

bility of probabilistic inference: If eventually Bayesian inference is as good as a point estimate even when the model is wrong, then what is the point of investing additional effort into the more challenging task of probabilistic inference?

### 1.2. Our Contributions and Related Work

Addressing model misspecification has long been an important direction in Bayesian statistics (Berger et al., 1994; Walker, 2013; Gelman & Yao, 2020), and it is beyond the scope of this review to cover all related approaches. When dealing with discrete models, it is well-known that ensemble methods like bagging (Domingos, 1997) and stacking (Le & Clarke, 2017; Yao et al., 2018; 2024) outperform full Bayesian model averaging under misspecification. Conceptually, the present paper shares the same spirit of predictive model averaging and can be viewed as its continuous generalization: instead of a discrete weight on a finite set of models, we find the predictively-optimal density in a continuous parameter space. Wang et al. (2019); Wang & Blei (2019) build the connection between variational methods and empirical Bayes. Louppe et al. (2019) use adversarial learning to minimize the divergence between the empirical distribution and the marginal distribution of data in SBI settings. Vandegar et al. (2021) approximate the likelihood of those simulators and optimize a neural empirical Bayes prior following the likelihood function. One interesting perspective is that the log score PVI performs neural empirical Bayes by minimizing the prior predictive loss. More related to our method, Masegosa (2020) develops a PAC-Bayes bound and Morningstar et al. (2022) establish PAC$^m$-Bayes. From a slightly different motivation to minimize the empirical risk gap and ensure the PAC generalization bound, their proposed PAC$^m$-Bayes solution can be viewed as an instance of our PVI with the special choice of log score, prior regularization, and one version of gradient evaluation. Another line of research focuses on direct loss minimization with expected log predictive density (Sheth & Khardon, 2017; 2020; Jankowiak et al., 2020) for Gaussian processes. Our work provides a practical inference pipeline to directly optimize the log predictive density and many other scoring rules for an arbitrary model. Concurrent with our work, Shen et al. (2025) derive maximum mean discrepancy (MMD)-based (Gretton et al., 2012) predictive inference, which corresponds to a similar predictive objective with MMD as the scoring rule. McLatchie et al. (2025) further develop related pseudo-posteriors for general scoring rules and study their theoretical properties. These works mainly use gradient-flow-based implementations. In contrast, we focus on tractable parametric variational implementations and provide a general recipe applicable to a wide range of proper scoring rules. One advantage of the parametric VI family is that it permits a spectrum of approximations: normalizing-flow PVI handles moderate-

---

**Algorithm 1** Variational Inference

**Require:** Initial parameter $\phi$, model $p(y|\theta)$, prior $p^{\mathrm{prior}}(\theta)$
1: **while** not converge **do**
2: $\quad \mathcal{L}_{\mathrm{VI}}(\phi) \leftarrow$
$\quad\quad \int_{\Theta} \sum_{i=1}^{n} \log p(y_i|\theta) q_\phi(\theta) d\theta - \mathrm{KL}(q_\phi(\theta)||p^{\mathrm{prior}}(\theta))$
3: $\quad \phi \leftarrow \phi + \eta \nabla_\phi \mathcal{L}_{\mathrm{VI}}(\phi)$
4: **end while**

---

**Algorithm 2** Predictive Variational Inference

**Require:** Initial parameter $\phi$, model $p(y|\theta)$, regularizer $r(\phi)$
1: **while** not converge **do**
2: $\quad \mathcal{L}_{\mathrm{PVI}}(\phi) \leftarrow$
$\quad\quad \sum_{i=1}^{n} S\left(\int_{\Theta} p(\cdot|\theta) q_\phi(\theta) d\theta, y_i\right) - \lambda r(\phi)$
3: $\quad \phi \leftarrow \phi + \eta \nabla_\phi \mathcal{L}_{\mathrm{PVI}}(\phi)$
4: **end while**

---

dimensional regressions well, while simpler families (e.g., mean-field Gaussian) further improve scalability in higher dimensions.

This work makes four primary contributions. First, we allow for a general scoring rule $S(\cdot, \cdot)$, which allows predictive optimality to be tuned to the scientific task at hand; in addition to the logarithmic score, we provide algorithms that produce calibrated predictive uncertainty (the interval score and CRPS) and the Brier score; the ability to handle divergences other than the logarithmic score is crucial, as CRPS can be implemented in a likelihood-free manner and therefore is amenable to likelihood-free simulation-based inference. Second, we derive gradients estimates for each divergence, including a novel unbiased gradient estimate in the logarithmic score using rejection sampling. Third, we allow for the PVI objective to be regularized either in the direction of the prior distribution or the posterior distribution, allowing us to smoothly interpolate between optimal predictive inference and fully-Bayesian inference. Last, we show how PVI can be used as a diagnostic tool and can motivate model expansion.

## 2. Predictive variational inference

### 2.1. Proper scoring rule of predictions

We evaluate probabilistic models by how the resulting posterior prediction fits the observed data. We measure such goodness-of-fit by scoring rules (for a complete review, see e.g., Gneiting & Raftery, 2007). We assume IID observations $(y_1, \ldots, y_n)$ on a sample space $\Omega$ from an unknown true data generating process $p_{\mathrm{true}}(y)$, $y \in \Omega$. Let $P$ be a probabilistic forecast for the next unseen data $\tilde{y}$. A scoring rule is an extended real-valued function $S(P, \tilde{y}) \in \overline{\mathbb{R}}$. The expected score measures the predictive performance of $P$ averaged over the true data generating process $\mathbb{E}[S(P, \tilde{y})] \coloneqq \int_\Omega S(P, \tilde{y}) dp_{\mathrm{true}}(\tilde{y})$, denoted by $S(P, p_{\mathrm{true}})$, and is often approximated by the empirical score $S(P, p_{\mathrm{true}}) \approx \frac{1}{n} \sum_{i=1}^{n} S(P, y_i)$. A scoring rule is said to be proper if the maximum expected score is achieved under the truth, i.e., $S(P, p_{\mathrm{true}}) \leq S(p_{\mathrm{true}}, p_{\mathrm{true}})$. A proper scoring rule corresponds to a divergence function in the $\Omega$ space between the forecast $P$ and the true data generating process $p_{\mathrm{true}}$, $D(P, p_{\mathrm{true}}) = S(p_{\mathrm{true}}, p_{\mathrm{true}}) - S(P, p_{\mathrm{true}})$.

The most common score is the log score, $S(P, \tilde{y}) = \log P(\tilde{y})$, which leads to the familiar KL divergence. Different scoring rules evaluate different aspects of the prediction, such as the predictive density, quantiles and coverage.

### 2.2. Optimizing the predictive performance

Consider the following inference task: We observe data $y = (y_1, \ldots, y_n)$, $y_i \in \Omega$, and assume $\{y_i\}$ are conditionally IID given covariates. Omitting covariates for brevity, we denote $p_{\mathrm{true}}(y_i)$ as the data generating process. We are given a (potentially wrong) model with a parameter vector $\theta \in \Theta$, a (potentially intractable) likelihood $p(y_i|\theta)$, and a prior $p^{\mathrm{prior}}(\theta)$. Let $p(\theta|y) \propto p^{\mathrm{prior}}(\theta) \prod_{i=1}^{n} p(y_i|\theta)$ be the exact Bayesian posterior density. Our inferential goal is a variational approximation $q_\phi(\theta)$, parameterized by a variational parameter $\phi \in \Phi$. Averaged over the inference $q_\phi$, the posterior predictive distribution of the next unseen data $\tilde{y}$ will be $q_\phi^Y(\tilde{y}) \coloneqq \int q_\phi(\theta) p(\tilde{y}|\theta) d\theta$. So far the setting is the same as standard VI. However, rather than approximating exact Bayes, we first pick a proper scoring rule $S$ on the outcome predictions, and then optimize $\phi$ such that the empirical scoring rule of the posterior predictive distribution is maximized. That is,

$$\max_{\phi \in \Phi} \left( \sum_{i=1}^{n} S\left( \int_{\Theta} p(\cdot|\theta) q_\phi(\theta) d\theta, y_i \right) - \lambda r(\phi) \right), \quad (3)$$

where $r(\phi)$ is an optional regularizer. If we choose $S$ to be the log score, PVI maximizes the usual log predictive density $\max_{\phi \in \Phi} \sum_{i=1}^{n} \log \int_{\Theta} p(y_i|\theta) q_\phi(\theta) d\theta$. Equivalently, as in (2), we can describe PVI as minimizing the divergence between the posterior predictive distribution, $q_\phi^Y(\tilde{y})$, and the true data generating process of the next unseen observation $p_{\mathrm{true}}(\tilde{y})$. Let $D$ be the corresponding divergence of the scoring rule $S$. Then up to a constant $C$, the entropy of $p_{\mathrm{true}}(\tilde{y})$, the first summation in the PVI objective (3) is the negative empirical version of this outcome-space divergence: $\lim_{n\to\infty} \sum_{i=1}^{n} S\left( q_\phi^Y(\cdot), y_i \right) / n - C = -D(q_\phi^Y(\tilde{y}) \ || \ p_{\mathrm{true}}(\tilde{y}))$. For example, the log score PVI minimizes the KL divergence to $p_{\mathrm{true}}(\tilde{y})$. We compare the procedure of VI and PVI in Algorithms 1 and 2.

## 2.3. Regularization

We include a tunable regularization term $-\lambda r(\phi)$ in the PVI objective (3). There are two reasons for this. First, compared to empirical risk minimization, it is a salient feature of Bayes to incorporate prior knowledge, which we will keep. Second, regularization improves the identifiability of $\phi$. The optimum is not necessarily unique in the PVI optimization of (3), and adding a regularizer stabilizes the location of the final solution. We consider two regularizers: regularizing to the prior, or to the posterior.

$$r^{\text{prior}}(\phi) = \text{KL}(q_\phi(\theta) \, || \, p^{\text{prior}}(\theta)). \qquad (4)$$

$$r^{\text{post}}(\phi) = \text{KL}(q_\phi(\theta) \, || \, p(\theta|y)). \qquad (5)$$

Regularization toward the prior (4) is familiar, as it appears in both standard VI and other generalized VI methods (see Section 5). The posterior-oriented regularization (5) is unique to PVI, and it is effectively the ELBO in standard VI. With this term, PVI can continuously interpolate between the pure prediction-optimization ($\lambda = 0$) and pure Bayes ($\lambda = \infty$), and thereby benefit from both finite-sample robustness and large-sample optimality guarantees. In most experiments, we keep $\lambda$ as a small fixed constant, though in principle it can also be tuned using cross-validation. See Supplement B.1 for more demonstrations of the effect of the regularization.

## 2.4. Asymptotics and hierarchical Bayes

Before we demonstrate the practical implementation of PVI (3) and various choices of $S$, we prove some appealing theoretical properties of the PVI optima. With enough samples and under certain assumptions, one would expect that the posterior prediction of the PVI solution converges to the best possible probabilistic forecast:

**Proposition 1** (informal). *Let the variational distribution $q_\phi(\theta)$ be parameterized by $\phi \in \Phi$. With any strictly proper score function $S$, an unknown true data generating process distribution $p_{\text{true}}(y)$, a likelihood model $p(y|\theta)$, and a size-$n$ sample $y_1, y_2, ..., y_n \sim p_{\text{true}}(\cdot)$, let $\phi_n$ be the solution of the PVI objective (3) with a continuous prior regularization on $\phi$. The predictive optimal variational parameter $\phi_0$ is defined as*

$$\phi_0 := \text{argmax}_{\phi \in \Phi} \, S\left(\int_\Theta p(\cdot|\theta)q_\phi(\theta)d\theta, \, p_{\text{true}}(\cdot)\right).$$

*Define $\ell(y, \phi) = S(\int p(\cdot|\theta) q_\phi(\theta) \, d\theta, y)$. If $\phi_0$ is unique, then under regularity conditions, we have (1) consistency: $\phi_n \xrightarrow{p} \phi_0$ as $n \to \infty$; (2) asymptotic normality: $\sqrt{n}(\phi_n - \phi_0) \xrightarrow{d} MVN(0, V)$ where $V$ is a non-singular matrix.*

The detailed proposition and proof are available in Supplement A.1. This consistency entails two corollaries.

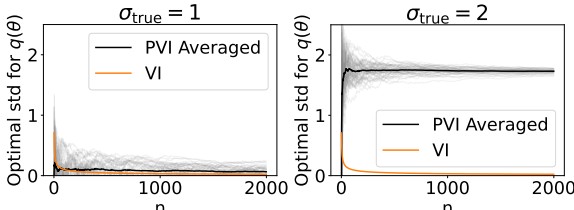

*Figure 2.* Optimal standard deviation of $q(\theta)$ with PVI or VI for the simple normal case from 50 simulations, with $\sigma_{\text{true}} = 1$ on the left and $\sigma_{\text{true}} = 2$ on the right.

**Corollary 2.** *If the model is well-specified, i.e., there exists a $\theta_0$ and $\phi^*$ such that $p(y|\theta_0) = p_{\text{true}}(y)$ and $q_{\phi^*}(\theta) = \delta_{\theta_0}(\theta)$, then under regularity conditions, $\phi_n \xrightarrow{p} \phi^*$.*

**Corollary 3.** *If the model is misspecified but there exists a $\phi^*$ such that $p_{\text{true}}(y) = \int p(y|\theta)q_{\phi^*}(\theta)d\theta$, then under regularity conditions, $\phi_n \xrightarrow{p} \phi^*$.*

The corollaries establish the backward compatibility of PVI when the variational family is flexible: If the model is well-specified, PVI concentrates on the true parameter, the same as regular Bayesian methods. However, if the model contains a parameter that varies in population, while the regular Bayes still concentrates, PVI asymptotically converges to the true population distribution of the parameter, not a point mass. We use a toy example to illustrate this difference.

**A normal example.** We model an IID dataset $\{y_i\}_{i=1}^n$ by a model $y|\theta \sim \text{normal}(\theta, 1)$ where the only parameter is the location $\theta$. The unknown true data generating procedure is $y_i \sim \text{normal}(0, \sigma_{\text{true}})$. The exact posterior is $p(\theta|y) = \text{normal}(\theta \mid \bar{y}, 1/\sqrt{n})$. We use PVI with a Gaussian variational family. (a) When $\sigma_{\text{true}} = 1$, the model is well-specified, and PVI solution $q(\theta)$ converges to the point mass at 0, same as the classical VI and Bayes. (b) When $\sigma_{\text{true}} = 2$, while the regular Bayes still concentrates to the point mass at 0, the PVI posterior converges to $\text{normal}(0, \sqrt{3})$, and stays there as $n$ goes to infinity regardless of the scoring rule, so that the posterior prediction matches the true data process despite the model being wrong. In general, unlike standard Bayesian inference, when the model is misspecified the PVI posterior uncertainty may not go to zero as the sample size grows to infinity, a reflection of the "epistemic humility". See Figure 2 for simulations.

**The uniqueness assumption.** Note in Proposition 1, we have an assumption that $\phi_0$ is unique. However, PVI is typically overparametrized: the PVI optimal solution may not be unique. Even in a normal model, $y \sim \text{normal}(\mu, \sigma)$, when both $\mu$ and $\sigma$ are allowed to be nondegenerate random variables, it is equivalent to distribute uncertainty either on the variance of $\mu$, or on the scale of $\sigma^2$. We are not worried about overparameterization for three reasons. (1) In practice,

we propose using regularization to break the ties, such that numerical optimization is often not an issue. (2) In our view, this overparameterization allows PVI to adjust for model misspecification to achieve better predictive performance. (3) Since we are using normalizing flows, even when the PVI solution is unique, the flow parameter will not be unique because a normalizing flow itself is typically not identifiable. Nevertheless, the overparameterization should not prevent PVI from being useful. We include the uniqueness assumption in Proposition 1 for theoretical interest. There is one special case when the uniqueness is typically satisfied–when the parametric model is 'correct' and the PVI optimum is a point delta function. Then our uniqueness assumption becomes the usual identifiable assumption of the original model, which is typically met. The main theoretical interest of Proposition 1 is to establish the consistency and convergence rate of PVI, such that it is 'backward compatible' with the regular Bayes when the model is correct.

### 2.5. Heterogeneity detection and implicit hierarchical Bayes

As implied by Corollary 2, a non-vanishing PVI posterior variance indicates model misspecification, which we can use as a diagnostic check[1]. In addition to a binary correct-or-wrong model check, monitoring the PVI posterior variance further indicates whether the parameters need to vary in population. With large enough $n$, if the variance of a variable $\mathrm{Var}(\theta_i) > \alpha$, where $\alpha$ is a chosen threshold, we may need to expand the model. We call this check "heterogeneity detection". We demonstrate how to use PVI to guide model expansion using an election example in Section 4.2. Another possible indicator of heterogeneity is the difference between the forecast scores with PVI and VI posteriors.

The parameter-may-vary-in-the-population perspective is the key motivation behind Bayesian hierarchical modeling (Gelman, 2006). Given any IID observational model $y \sim p(y|\theta)$, we distinguish between three views of inference:

1. *Complete-pooling* view: We perform standard Bayesian inference $\theta|y$. It is straightforward but more prone to model misspecification.

2. *Full-hierarchical* view: Any parameter is assumed to vary in the population and we aim for a fully individualized inference. That is, we expand the model as $y_i \sim p(y|\theta_i)$, and infer $\theta|y$ where $\theta = (\theta_1, \ldots, \theta_n)$. This hierarchical expansion makes the model more robust (Wang & Blei, 2018), with the cost of augmenting each parameter $\theta$ by $n$ copies $\theta_1, \ldots, \theta_n$, which quickly becomes computationally prohibitive when $n$ is big. Moreover, as each parameter $\theta_i$ only links to

one data point $y_i$, this individualized inference is typically noisy and relies on the hyper-prior, which itself is a parametric family and may be misspecified.

3. *Implicit-hierarchical* view: We implicitly adopt the view of $y_i \sim p(y_i|\theta_i)$, but we marginalize out individual $\theta_i$ and only infer their population distribution $\theta_1, \theta_2, \ldots, \theta_n \sim \pi_{\mathrm{pop}}(\theta)$. In many cases, the primary inference interest lies in this population distribution, such as understanding population-level properties or making predictions for new data points.

PVI is an implicit hierarchical expansion. Under the condition of Corollary 3, $q_\phi(\theta)$ converges to $\pi_{\mathrm{pop}}(\theta)$, the above-mentioned population distribution of parameters, offering an automated and computationally fast way to perform hierarchical expansion for any model, and adapting to any scoring rule.

## 3. Implementation and gradient evaluation

PVI is a natural idea, but its optimization is nontrivial. This section provides practical stochastic gradient algorithms to implement PVI (3) on four concrete scoring rules $S$: the logarithmic, quadratic, interval and continuous ranked probability score.

The PVI objective (3) contains two terms. The regularization term is the usual ELBO (posterior regularizer) or part of the ELBO (prior regularizer), hence its gradient with respect to the variational parameter $\phi$ is straightforward. The first term in (3) is a summation: $R(\phi) := \sum_{i=1}^n S\left(\int p(\cdot|\theta)q_\phi(\theta)d\theta, y_i\right)$. In stochastic optimization, in each iteration we sample a mini-batch of $\theta_1, \ldots, \theta_M$ draws from the variational density $q_\phi(\theta)$. To iteratively update $\phi$, we need to evaluate the gradient $\frac{d}{d\phi}R(\phi)$. As an overview, we summarize the evaluation of $R(\phi)$ and the gradient $\frac{d}{d\phi}R(\phi)$ in Table 1. We defer theoretical claims and algorithm details to the Supplement A.

**The logarithmic score with unbiased gradient** $S(Q, y) = \log Q(y)$ is the default score in Bayesian inference. The log-score PVI equipped with the prior regularization and the asymptotically unbiased gradient (a) coincides with the PAC$^m$-Bayes (Morningstar et al., 2022). Because of the exchange in the order of the integral and log, the objective $R(\phi)$ contains the logarithm of a stochastic summation, making its gradient evaluation nontrivial. Table 1 shows two gradient estimates $\frac{d}{d\phi}R(\phi)$. The first estimate approximates the integral by reparameterization, and we show it is asymptotically unbiased when the batch size $M$ is big. With finite batch size, however, the gradient estimator is biased. In the extreme case of $M = 1$, the bias is so severe that the gradient reduces to that of standard VI. To correct for this finite-batch bias, we further develop a sec-

---

[1] As a caveat, the uniqueness assumption in Proposition 1 may not always hold.

| Rule | Objective to maximize | Gradient estimator |
|------|----------------------|--------------------|
| Logarithmic score | $\sum_{i=1}^{n} \log \int_{\Theta} p(y_i\|\theta) q_\phi(\theta) d\theta$ | $\boxed{a}\ \sum_{i=1}^{n} \left( \sum_{j=1}^{M} \frac{d}{d\phi} p(y_i\|T_\phi(u_j)) \right) / \left( \sum_{j=1}^{M} p(y_i\|T_\phi(u_j)) \right)$ 
 $\boxed{b}\ \sum_{i=1}^{n} \frac{\sum_{j=1}^{M} \left( \mathbb{1}(t_j < p(y_i\|\theta_j)/C) \frac{\partial}{\partial\phi} \log q_\phi(\theta_j) \right)}{\sum_{j=1}^{M} \mathbb{1}(t_j < p(y_i\|\theta_j)/C)}$ |
| Quadratic score | $\sum_{i=1}^{n} 2 \int_{\Theta} f(\theta, y_i) q_\phi(\theta) d\theta$ 
 $- \sum_{j=1}^{I} \left( \int_{\Theta} f(\theta, j) q_\phi(\theta) d\theta \right)^2$ | $2 \sum_{i=1}^{n} \frac{1}{M} \sum_{j=1}^{M} \frac{d}{d\phi} f(T_\phi(u_j), y_i)$ 
 $- \sum_{j=1}^{I} \left( \frac{1}{M} \sum_{k=1}^{M} f(T_\phi(u_k), j) \right) \left( \frac{1}{M} \sum_{k=1}^{M} \frac{df(T_\phi(u_k), j)}{d\phi} \right)$ |
| Interval score | $- \sum_{i=1}^{n} (U_\alpha - L_\alpha) + \frac{2}{\alpha}(L_\alpha - y_i)\mathbb{I}(y_i < L_\alpha)$ 
 $+ \frac{2}{\alpha}(y_i - U_\alpha)\mathbb{I}(y_i > U_\alpha)$ | $\sum_{i=1}^{n} \frac{\partial \hat{U}_\alpha}{\partial\phi} \left( \frac{2}{\alpha}\mathbb{I}(y_i > \hat{U}_\alpha) - 1 \right) - \frac{\partial \hat{L}_\alpha}{\partial\phi} \left( \frac{2}{\alpha}\mathbb{I}(y_i < \hat{L}_\alpha) - 1 \right)$ |
| CRPS | $- \sum_{i=1}^{n} \mathbb{E}_{y^{\text{sim}} \sim \int_{\Theta} p(y\|\theta) q_\phi(\theta) d\theta} \left[ \|y^{\text{sim}} - y_i\| \right]$ 
 $+ \frac{1}{2} \mathbb{E}_{y_1^{\text{sim}}, y_2^{\text{sim}} \sim \int_{\Theta} p(y\|\theta) q_\phi(\theta) d\theta} \left[ \|y_1^{\text{sim}} - y_2^{\text{sim}}\| \right]$ | $- \frac{1}{2M} \sum_{i=1}^{n} \sum_{m=1}^{2M} \left( g_m \cdot \text{sign}(y_m^{\text{sim}} - y_i) \right)$ 
 $+ \frac{1}{2M} \sum_{m=1}^{M} \left( (g_m - g_{m+M}) \cdot \text{sign}(y_m^{\text{sim}} - y_{m+M}^{\text{sim}}) \right)$ |

*Table 1.* How to compute the objectives and their gradients for four scoring rules. Here, $\theta = T_\phi(u)$, $u \sim q(\cdot)$ is a reparameterization such that $\theta \sim q_\phi(\cdot)$. We generate $2M$ independent observations $y_1^{\text{sim}}, \ldots, y_{2M}^{\text{sim}}$ for CRPS and denote $g_m = \frac{d}{d\phi} y_m^{\text{sim}}$ for $m = 1, \ldots, 2M$. We estimate sample quantiles $\hat{L}_\alpha$ and $\hat{U}_\alpha$ for the interval score from simulated observations.

ond novel gradient estimate based on rejection-sampling. For this gradient, at each SGD iteration given a $\phi$, we draw $M$ IID copies $\theta_1, \ldots, \theta_M$ from $q_\phi(\theta)$, and $M$ IID copies $t_1, \ldots, t_M$ from Uniform$(0,1)$ distribution. For each $j$, we compute $p(y_i\|\theta_j)/C$. If $t_j < p(y_i\|\theta_j)/C$ then we accept the draw $\theta_j$ for $y_i$. If at least one draw is accepted we approximate $g_i^{\log}(\phi) = \frac{d}{d\phi} \log \int p(y_i\|\theta) q_\phi(\theta) d\theta$ by

$$g_i^{\text{RS}} := \frac{\sum_{j=1}^{M} \left( \mathbb{1}(t_j < p(y_i\|\theta_j)/C) \frac{\partial}{\partial\phi} \log q_\phi(\theta_j) \right)}{\sum_{j=1}^{M} \mathbb{1}(t_j < p(y_i\|\theta_j)/C)}. \quad (6)$$

We show that this novel gradient estimate (9) is always unbiased, regardless of the batch size, and hence guarantees the SGD convergence under regular conditions.

**Proposition 4.** *For any $M \geq 1$, $\mathbb{E}_{\theta,t}[g_i^{\text{RS}}] = g_i^{\log}(\phi)$.*

See Supplement A.2 for proofs and more details.

**The quadratic score** or equivalently the Brier score (Brier, 1950) is useful to examine predictions on categorical-valued outcomes, $y_i \in \Omega = \{1, 2, ..., I\}$. It is sometimes more sensitive than the log score for prediction evaluations (Selten, 1998). To run PVI, we count the occurrences in the data: category $i$ has $n_i$ occurrences. Denote the conditional predictive mass function by $f(\theta, y_i) = \Pr(Y = y_i\|\Theta = \theta)$; then, the posterior predictive distribution is $\Pr(Y = y_i) = \int_{\Theta} f(\theta, y_i) q_\phi(\theta) d\theta$. We present the objective and its gradient estimator in the second row of Table 1. In the supplement, we show that this gradient evaluation converges in probability and leads to a valid PVI optimization.

**The interval score** (IS, Dunsmore, 1968; Winkler & Murphy, 1968) measures the sharpness and calibration of the

predictive intervals given true data. Suppose $y \in \mathbb{R}$. Under the significance level $\alpha$, the probabilistic forecast $Q$ corresponds to the prediction interval $[L_\alpha, U_\alpha]$. The objective of the interval score is in the third row of Table 1. In PVI, we need to optimize the parameter $\phi$ by minimizing the interval score of the predictive model $Q_\phi(y) = \int_{\Theta} p(y\|\theta) q_\phi(\theta) d\theta$. However, it may be hard to derive $L_\alpha$ and $U_\alpha$ analytically. Instead, we generate $M$ samples, estimate the sample quantiles $\hat{L}_\alpha$ and $\hat{U}_\alpha$, and compute the IS using the quantile estimates. During optimization, we use the gradients of estimated quantiles from autodiff, $\frac{\partial \hat{L}_\alpha}{\partial\phi}$ and $\frac{\partial \hat{U}_\alpha}{\partial\phi}$. In the supplement, we show these gradient estimators are convergent.

**The continuous ranked probability score** (CRPS, Matheson & Winkler, 1976) measures the $L^2$ distance between the cumulative density functions (CDF) of the prediction and true data-generating distribution. It could be interpreted as the combination of IS across all significance levels (Gneiting & Ranjan, 2011). To start, assume the observation $y_i \in \mathbb{R}$ is a scalar. Typically neither the true nor the predicted CDF is of a closed-form, and we evaluate the CRPS via the simulations from the posterior predictions: all we need is to draw $\theta$ from $q_\phi(\theta)$, and then $y^{\text{sim}}$ from $p(y\|\theta)$ in sequence. The fourth row of Table 1 presents the CRPS objectives, and its gradient. Our estimates only require simulations $y^{\text{sim}}$ from the posterior predictive, not the density, making it readily usable for simulation-based inference (SBI). We prove this gradient is always unbiased as long as $y^{\text{sim}}$ can be differentiated through, leading to a stronger convergence guarantee for the optimization.

Although the original CRPS is designed for one-dimensional outcomes, it naturally extends to higher dimensional out-

comes by replacing the scalar absolute value $|\cdot|$ in the objective with a non-negative, continuous negative-definite kernel function (for details, see Gneiting & Raftery, 2007). In our experiment when the outcome $y_i$ is high-dimensional microscopy images, we simply plug in the $L^2$ distance between observed $y_i$ and simulations $y^{\text{sim}}$ in the CRPS and gradient evaluation, which is still a valid proper scoring rule.

**PVI in likelihood-free inference.** Many recent scientific applications involve intractable likelihoods: the sampling distribution $p(y|\theta)$ may contain a complex simulator, such that we cannot evaluate the conditional density $p(y|\theta)$ but still can draw simulations $y^{\text{sim}}$ from it. Simulation-based inference (SBI, Cranmer et al., 2020) is a successful tool for likelihood-free tasks, but the best an SBI algorithm can achieve is to be faithful to the Bayesian posterior, which may suffer from model-misspecification (e.g., Cannon et al., 2022). In contrast, because we have developed the general PVI method incorporating an arbitrary scoring rule, we handle likelihood-exact and -free inference in a unified framework. In particular, our proposed PVI with CRPS is well-suited for such likelihood-free applications: All we need is a continuous outcome space ($y \sim \mathbb{R}^d$) and a differentiable simulator. For all the reasons given above, PVI is more robust against model misspecification and better fits the true data process. In our later cryoEM experiment in Section 4.3, we use PVI-CRPS to learn the population distribution of frozen molecules with a simulator. Here, the PVI-learned $q_\phi(\cdot)$ is not merely a prediction-oriented augmentation, but a physically meaningful quantity.

# 4. Experiments

We present three examples. We show how PVI is used as a tool for detecting model misspecification in a golf putting model. In the election example, we use PVI as a heterogeneity detection and a guidance for model expansion. In a cryoEM example, we apply PVI with CRPS to infer protein structures from an intractable likelihood. Additional experiments are in Supplement B. In Section B.5 we include a benchmark on 7 `posteriordb` (Magnusson et al., 2024) models to compare different scoring rules, where we treat the regularization strength $\lambda$ and the choice of whether to regularize towards the prior or posterior as tuning parameters, and we use cross-validation to select them. The code is available at https://github.com/lll6924/pvi.git.

## 4.1. Model misspecification detection for golf putting

Gelman & Nolan (2002) modeled the proportion of successful putts from professional golfers as a function of distance from the hole. There were two models for the problem: the logistic regression model and the geometric model. We plot

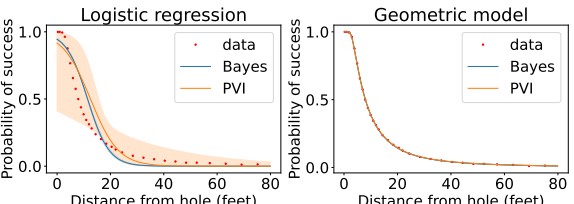

*Figure 3.* Posterior predictive distribution of two different models of the golf putting problem, with either the Bayesian posterior or the PVI posterior.

the posterior predictive distribution of both models with either the Bayesian or PVI posterior in Figure 3. Note that logistic regression is misspecified so the Bayesian posterior struggles to fit the curve and selects a narrow and incorrect prediction. But such misspecification is addressed by PVI with a wider posterior. The geometric model fits the data well, and both the Bayesian and PVI posterior select the correct parameters. This indicates that PVI can be used as a tool for misspecification detection. When a wide posterior is observed, we should modify the model for better fit. We will demonstrate further with the election model next.

## 4.2. Model expansion in U.S. election analysis

In the next application in survey sampling, we use PVI to both guide the hierarchical model building, and improve the within-stratum predictive performance, which is crucial in multilevel regression and post-stratification (MRP). As previously studied by Ghitza & Gelman (2013), we analyze a U.S. presidential election example, using the Current Population Survey's (CPS) post-election voting and registration supplement in the Year 2000 ($N =$68,643 respondents) to fit a binomial regression model that predicts voter turnouts given a respondent's state ($\{1, 2, ..., 51\}$ including D.C.), ethnicity group ($\{1, 2, 3, 4\}$) and income level ($\{1, 2, ..., 5\}$). Bayesian multilevel modeling typically starts by modeling the marginal state-, ethnicity- and income-effects. Next, the income effect may vary by state, so it makes sense to model the two-way interactions, and then the three-way interactions. Hence, even in a simple binomial regression with full interactions, we have at least $51 \times 5 \times 4 =$1,020 coefficient parameters. But it is still not saturated: there are still many unobserved confounders and hence this state$\times$income$\times$ethnicity effect may vary in the population, and an individualized fully-hierarchical model expansion (Section 2.4) would contain 1,020$\times$68,643 = 70 million parameters. We need some guidance on how deep the interactions we want to include in the model.

To test how PVI helps with model checking and further guides iterative model building, we first generate synthetic data so that we can compare the inference with a ground truth. In the survey data, in the $n$-th cell defined by the state $i_n$, ethnicity group $j_n$, and income level $x_n$, out of $N_n$ total

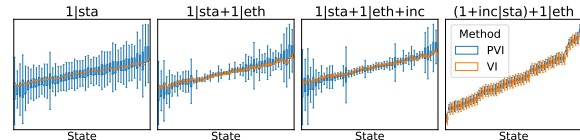

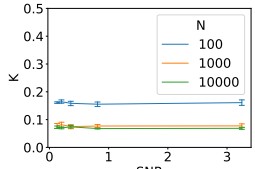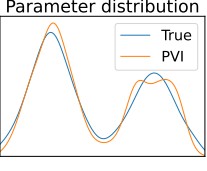

*Figure 4.* Distribution of inferred logit across states with $x = 1$ for ethnic group 1 from four different voting models. To focus on the variability across states, we set $\beta_{2,1}$ to its sample mean.

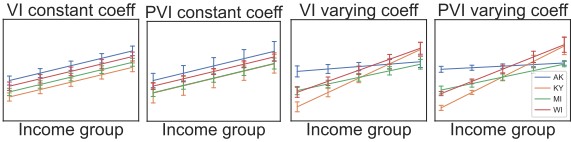

*Figure 5.* Regression logits of four states learned from the Current Population Survey's (CPS) post-election voting and registration supplement in 2000. Unrelated parameters are replaced by their sample means. Two models are compared: one with a constant coefficient, and one with varying coefficients among states.

respondents, $y_n$ of them are positive outcomes (1 = turnout). We simulate a dataset from a generating process:

$$y_n \sim \text{Binomial}(N_n, \text{Logit}^{-1}(\beta_{1,i_n} + \beta_{2,j_n} + \beta_{3,i_n} x_n)),$$

where $\boldsymbol{\beta}_1 \in \mathbb{R}^{51}$, $\boldsymbol{\beta}_2 \in \mathbb{R}^4$, $\boldsymbol{\beta}_3 \in \mathbb{R}^{51}$. We then run inference (our PVI and classical VI) on the well-specified model and three incomplete/misspecified models

$$y_n \sim \text{Binomial}(N_n, \text{Logit}^{-1}(\beta_{1,i_n})),$$
$$y_n \sim \text{Binomial}(N_n, \text{Logit}^{-1}(\beta_{1,i_n} + \beta_{2,j_n})),$$
$$y_n \sim \text{Binomial}(N_n, \text{Logit}^{-1}(\beta_{1,i_n} + \beta_{2,j_n} + \beta_3 x_n)).$$

Figure 4 demonstrates the inferred logits across states with different models. Despite models being misspecified, VI always concentrates, while our PVI correctly picks a wider variational distribution, which we regard as evidence of model misspecification. This illustrates PVI's value for model checking; the closer a statistical model is to the truth, the less posterior PVI variability is observed. In the last subfigure, the PVI posterior successfully concentrates when the model is correct, to the same parameters as VI.

Next, in the real data analysis, we fit two models using both PVI and VI: a model with (a) a constant coefficient across states, and (b) varying coefficients across states. We plot the logits of four different states in Figure 5 to demonstrate the behaviors of PVI. PVI on the constant coefficient model reveals that the posterior variances of some states (AK, KY) are higher than other states, suggesting that we should expand the model to allow varying coefficients in these states, which agrees with the conclusion of Ghitza & Gelman (2013). In the varying coefficients model, the two states indeed have different slopes. In contrast, VI assigns equal variances to all states due to the model restriction.

*Figure 6.* Parameter inference for cryoEM with PVI. In the first panel, the y-axis reports the gap $K$ between the inferred distribution and the true distribution with varying SNR and sample size, where the $K$ is computed from a two-sample KS test. The second figure demonstrates that the PVI inferred posterior distribution accurately recovers the true distribution in a realistic setting when $N = 10,000$ and SNR $\approx 0.13$.

### 4.3. Likelihood-free cryoEM inference

Inferring the structures of proteins is a challenging problem for computational biology. With the technique of cryo-electron microscopy (cryoEM, Nogales, 2016; Singer & Sigworth, 2020), scientists obtain a large collection of 2D protein images. Recently, Dingeldein et al. (2024) used SBI to analyze cryoEM images. The parameter of interest $\theta$ represents the protein structure, the sampling model of images follows $\mathbf{y} = f(\theta, \boldsymbol{\epsilon})$, where $\boldsymbol{\epsilon} \sim p^{\text{noise}}(\cdot)$ represents other nuisance parameters (rotations, translations, and measurement errors). The simulator is differentiable, but the likelihood $p(\mathbf{y}|\theta)$ is intractable because it is difficult to integrate all parameters. Moreover, the protein structure $\theta$ is not one single fixed truth across all images: the proteins were distributed via their free energy distribution when they were frozen. Due to computational challenges, this protein dynamic is not fully addressed in past works, which only attempt to approximate the exact but overconfident Bayes posterior $\theta|\mathbf{y}$ even when the image sample size $N$ is big. We use CRPS-equipped PVI to find an optimal $\phi$, such that the simulations from $\int_{\Theta} p(y|\theta)q_{\phi}(\theta)d\theta$ match the observed images as close as possible under the CRPS metric. This approach is computationally efficient as it does not need likelihood evaluations or expensive prior simulations. The learning of the $\theta$-distribution is effectively nonparametric as we use a normalizing flow.

We work with a realistic simulator of heat shock proteins (HSP90, Whitesell & Lindquist, 2005). An HSP90 protein contains two chains where the opening $\theta$ controls the conformation. We choose a parameter distribution with two modes and generate a synthetic dataset of noisy images. In the simulator, the noise includes a normal error and a random defocus for the contrast transfer function (CTF) of the lens, making the signal-to-noise ratio (SNR) of images very low. Figure 1 shows that the variational posterior $q_{\phi}(\cdot)$ learned from PVI with CRPS quickly matches the population ground-truth openings of the protein population, while the Bayes posterior ignores heterogeneity and collapses to point estimates. Figure 6 visualizes the accuracy of the PVI

posterior as a function of SNR and sample size: the PVI posterior converges to the target even with a low SNR.

## 5. Discussion

**Generalized VI and generalized Bayes.** It is not a new idea to reframe Bayesian inference as an infinite dimensional optimization. As mathematically equivalent to (1), Zellner (1988) stated that the minimizer of $\mathbb{E}_{\theta \sim q(\cdot)} \sum_{i=1}^{n} [-\log p(y_i|\theta)] + \mathrm{KL}(q \,||\, p^{\mathrm{prior}})$ is the classical Bayesian posterior $q(\cdot) = p(\cdot|y)$, while the vanilla VI is the solution of the constrained optimization. Motivated by this view, many variations of Bayes have emerged. Knoblauch et al. (2022) define a "generalized VI" framework (rule-of-three), $\min_{q \in \mathcal{F}} \mathbb{E}_{\theta \sim q(\cdot)} \left( \sum_{i=1}^{n} l(y_i, \theta) \right) + D(q \,||\, p^{\mathrm{prior}})$, where the user can choose a $(y \times \theta)$-space loss function $l(y_i, \theta)$, a $\theta$-space divergence $D$, and a feasible region $\mathcal{F}$, and this framework includes many other generalized-VIs as special cases such as decision-theory-based-VI (Lacoste-Julien et al., 2011; Kuśmierczyk et al., 2019), and Gibbs-VI (Alquier et al., 2016). In the same vein, Bissiri et al. (2016) study a generalized Bayes framework: given a $(y \times \theta)$-space loss function $l(y_i, \theta)$, an unnormalized posterior is defined by $\exp\left[ -\sum_{i=1}^{n} l(\theta, y_i) \right] p^{\mathrm{prior}}(\theta)$, a generalization of the PAC- and Gibbs-posterior (Zhang, 2006; Jiang & Tanner, 2008). Grünwald & Van Ommen (2017) introduce a learning rate $\eta$ to the Bayes model, which controls how close the posterior should be to the prior. When the likelihood is intractable, Matsubara et al. (2022) use Stein discrepancy as a loss function in generalized Bayes.

Our PVI framework is spiritually different from existing generalized-Bayes and -VI procedures. To see their difference, take the default loss function, the negative log likelihood, $l(y, \theta) = -\log p(y|\theta)$, and $D$ to be the KL divergence, then both generalized VI and generalized Bayes degenerate to the vanilla Bayes posterior. In contrast, the PVI posterior under the default log score is generally not the same, even asymptotically, as the vanilla Bayes posterior. Even with the same log score, PVI and the generalized VI differ in the position of $\log$ and the integral — PVI evaluates the global predictive performance of the posterior-averaged predictive distribution, while the generalized VI assesses the posterior-averaged local predictive performance.

**Limitations and future directions.** In this paper, we performed minimal tuning of PVI for a fair comparison. There is large room for fine-tuning (tuning $\lambda$, interpolating between the prior and posterior regularization, choice of scoring rules and combinations, variance reduction of the stochastic gradient, etc.). Throughout the paper, we have relied on the assumption that observations are conditionally IID given covariates, or at least exchangeable, which is a limitation compared with the regular Bayes. This IID requirement is shared by any scoring-rule-based model evaluation, generalized VI, and any loss-function-based learning algorithm. We can extend our current IID PVI to complex data structures; it requires defining internal replications and is beyond this paper.

Theoretically we justify PVI via asymptotic optimality. It is useful to further study its finite sample rate, and compare the convergence rate under various scoring rules. Unlike regular Bayes which may be overconfident, the parameter uncertainty in PVI can grow as a function of sample size (Figure 2), and may not shrink to zero, suggests the potential of turning PVI into a new uncertainty measurement.

PVI is robust to model misspecification and flexible for prediction, but these benefits come with costs. First, as illustrated in the Gaussian example in Figure 2, when the model is correctly specified, the PVI posterior still concentrates at the correct point mass asymptotically, but its contraction rate is slower than the usual parametric rate. This slower rate agrees with the theoretical analysis in concurrent work by McLatchie et al. (2025). Second, the PVI learning task (3) in probability space is harder than finite-dimensional parameter learning, limiting scalability in modern high-dimensional models. We view this increased computational burden as a tradeoff between a *complex model* and *complex inference*. While classical machine learning often focuses on making the model deeper, PVI instead pushes the complexity of inference. For scalable implementation, a promising future direction is to study the allocation between model and inference complexity, and to develop structured variational families and customized optimization schemes for PVI.

In this paper, we demonstrate practical implementations of PVI under several scoring rules. On one hand, practitioners may have their own decision-theoretic scoring rules, and PVI can accommodate these choices. On the other hand, PVI provides a unified framework for comparing the efficiency and robustness of different scoring rules when used for inference, which we leave for future research.

## Acknowledgment

The authors thank Pilar Cossio, Luke Evans, and Erik Thiede for help with cryoEM experiments. This work was partially completed while YY and JL were at the Flatiron Institute. ARL is supported by NSF grant DMS-2144933.

## Impact Statement

This paper presents methodological work whose goal is to advance the general area of probabilistic inference. There may be potential societal consequences of our work when it is applied in specific domains, none of which we feel must be specifically highlighted here.

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

# Appendices to "Predictive Variational Inference"

In the appendices, we provide proofs and additional discussion of the theoretical claims. We also provide details of our numerical experiments.

## A. Proofs

### A.1. Proof of Proposition 1

We restate Proposition 1 formally and prove the properties.

**Proposition 1** (formal). *Suppose the variational distribution $q_\phi(\theta)$ is parameterized by $\phi \in \Phi$, where $\Phi$ is compact. With any strictly proper score function $S$, an unknown true data generating process distribution $p_{\text{true}}(y)$, a likelihood model $p(y|\theta)$, and a size-$n$ sample $y_1, y_2, ..., y_n \sim p_{\text{true}}(\cdot)$, let $\phi_n$ be the solution of the PVI objective with a continuous prior regularization on $\phi$*

$$\phi_n = \text{argmax}_{\phi \in \Phi} \left( \sum_{i=1}^{n} S\left( \int p(\cdot|\theta) q_\phi(\theta) d\theta, y_i \right) - \lambda r^{\text{prior}}(\phi) \right). \tag{7}$$

*The predictively optimal variational parameter $\phi_0$ is defined as*

$$\phi_0 := \text{argmax}_{\phi \in \Phi} S\left( \int p(\cdot|\theta) q_\phi(\theta) d\theta, p_{\text{true}}(\cdot) \right).$$

*Define $\ell(y, \phi) = S(\int p(\cdot|\theta) q_\phi(\theta) d\theta, y)$, then we have (1) consistency: if $\mathbb{E}_{y \sim p_{\text{true}}} \left[ \sup_{\phi \in \Phi} |\ell(y, \phi)| \right] < \infty$, $\phi_0$ is unique, and $\ell(y_i, \phi)$ is continuous at each $\phi \in \Phi$, then $\phi_n \xrightarrow{p} \phi_0$ as $n \to \infty$; (2) asymptotic normality: if additionally, $\phi_0 \in \text{interior}(\Phi)$, $\ell(y_i, \phi)$ is twice continuously differentiable, $\sqrt{n} \nabla_\phi \frac{1}{n} \sum_{i=1}^{n} \ell(y_i, \phi_0) \xrightarrow{d} MVN(0, \Sigma)$, and $\sup_\phi \|\nabla_\phi^2 \frac{1}{n} \sum_{i=1}^{n} \ell(y_i, \phi) - H(\phi)\| \xrightarrow{p} 0$ for a continuous $H(\phi)$, and let $H = H(\phi_0)$ which is not singular, then $\sqrt{n}(\phi_n - \phi_0) \xrightarrow{d} MVN(0, H^{-1} \Sigma H^{-1})$.*

*Proof.* The proof is technically the same as the convergence and asymptotic normality of MLE, with additional consideration of the regularization. There exist other sets of assumptions that lead to the same results. To simplify the notations here, we define

$$T(\phi) = S\left( \int p(\cdot|\theta) q_\phi(\theta) d\theta, p_{\text{true}}(\cdot) \right),$$

$$T_n'(\phi) = \frac{1}{n} \sum_{i=1}^{n} \ell(y_i, \phi),$$

$$d(y) = \sup_{\phi \in \Phi} |\ell(y, \phi)|.$$

Then $\phi_0 = \text{argmax}_{\phi \in \Phi} T(\phi)$. Our assumption says that $\ell(y_i, \phi)$ is continuous at each $\phi$ and $|\ell(y, \phi)| \leq d(y)$ for $\mathbb{E}_{y \sim p_{\text{true}}}[d(y)] < \infty$. By the dominated convergence theorem, $T(\phi) = \mathbb{E}_{y \sim p_{\text{true}}}[\ell(y, \phi)]$ is continuous on $\phi$. Using uniform law of large numbers,

$$\sup_{\phi \in \Phi} |T_n'(\phi) - T(\phi)| \xrightarrow{p} 0. \tag{8}$$

This basically implies the convergence result when $\lambda = 0$. Next, we define

$$T_n(\phi) = \frac{1}{n} \sum_{i=1}^{n} \ell(y_i, \phi) - \frac{\lambda}{n} r^{\text{prior}}(\phi),$$

such that $\phi_n = \text{argmax}_{\phi \in \Phi} T_n(\phi)$. As $\Phi$ is compact, there exists $U$ such that $0 \leq r^{\text{prior}}(\phi) \leq U$. Hence $\sup_{\phi \in \Phi} |T_n(\phi) - T_n'(\phi)| \to 0$. Along with (8), we have that

$$\sup_{\phi \in \Phi} |T_n(\phi) - T(\phi)| \xrightarrow{p} 0.$$

With Theorem 2.1 in Newey & McFadden (1994), we have $\phi_n \overset{p}{\to} \phi_0$.

For asymptotic normality, note that $\nabla_\phi \frac{\lambda}{n} r^{\text{prior}}(\phi) = \mathcal{O}(1/n)$ and $\nabla_\phi^2 \frac{\lambda}{n} r^{\text{prior}}(\phi) \to 0$, so we further have $\sqrt{n} \nabla_\phi T_n(\phi) \overset{d}{\to}$ $\text{MVN}(0, \Sigma)$, and $\sup_\phi \|\nabla_\phi^2 T_n(\phi) - H(\phi)\| \overset{p}{\to} 0$. With Theorem 3.1 in Newey & McFadden (1994), we also have the asymptotic normality. $\square$

### A.2. Unbiased rejection sampling for the Logarithmic score gradient

We propose a rejection-sampling-based gradient estimate for each $g_i^{\log}(\phi) = \frac{d}{d\phi} \log \int p(y_i|\theta) q_\phi(\theta) d\theta$ under log score. It is particularly simple when the pointwise likelihood $p(y_i|\theta)$ is bounded. That is, there exists a known scalar constant $C > 0$ such that $p(y_i|\theta) < C$ for all $i$ and $\theta \in \Theta$. We will discuss this boundedness condition later.

At each SGD iteration given a $\phi$, we draw $M$ IID copies $\theta_1, \ldots, \theta_M$ from $q_\phi(\theta)$, and $M$ IID copies $t_1, \ldots, t_M$ from Uniform(0,1) distribution. Pointwise, we compute $p(y_i|\theta)/C$, if $t_j < p(y_i|\theta_j)/C$ then we accept the draw $\theta_j$. If at least one draw is accepted, we approximate the gradient $g_i^{\log}(\phi)$ by

$$g_i^{\text{RS}} := \frac{\sum_{j=1}^M \left( \mathbb{1}(t_j < p(y_i|\theta_j)/C) \frac{\partial}{\partial \phi} \log q_\phi(\theta_j) \right)}{\sum_{j=1}^M \mathbb{1}(t_j < p(y_i|\theta_j)/C)}. \tag{9}$$

If none of the draws are accepted, we resample $\theta_1, \ldots, \theta_M$ and $t_1, \ldots, t_M$ until we have an estimator. Finally we compute $g^{\text{RS}} = \sum_{i=1}^n g_i^{\text{RS}}$.

Shifting from importance sampling to rejection sampling seems a small step change, but it ensures an unbiased gradient estimate even with one Monte Carlo draw, i.e.,

**Proposition 4.** *For any sample size $M \geq 1$, $\mathbb{E}_{\theta,t}[g_i^{\text{RS}}] = g_i^{\log}(\phi)$.*

*Proof.* Without loss of generality, assume $C = 1$. We first show that each $g_i^{\text{RS}}$ is an unbiased estimator for $g_i^{\log}(\phi)$. We consider a subsequence of the auxiliary samples $t_{n_1}, t_{n_2}, \ldots, t_{n_k}$, where $1 \leq n_1 < n_2 < \ldots < n_k \leq M$ and $N = \{n_1, \ldots, n_k\}$ is the set of indices, and condition on the case where their draws are accepted but others are rejected. Denote the predictive distribution by $q_\phi^Y(y_i) = \int p(y_i|\theta) q_\phi(\theta) d\theta$. Then we have

$$\mathbb{E}[g_i^{\text{RS}}|\text{accept } t_{n_1}, \ldots, t_{n_k}]$$

$$= \frac{\mathbb{E}_{\text{accept } t_{n_1}, \ldots, t_{n_k}}[g_i^{\text{RS}}]}{P(\text{accept } t_{n_1}, \ldots, t_{n_k})}$$

$$= \frac{\int \cdots \int \prod_{m=1}^M q_\phi(\theta_m) \prod_{j=1}^k p(y_i|\theta_{n_j}) \prod_{j \notin N}(1 - p(y_i|\theta_j)) \frac{1}{k} \sum_{j=1}^k \frac{\partial}{\partial \phi} \log q_\phi(\theta_{n_j}) d\Theta}{\int \cdots \int \prod_{m=1}^M q_\phi(\theta_m) \prod_{j=1}^k p(y_i|\theta_{n_j}) \prod_{j \notin N}(1 - p(y_i|\theta_j)) d\Theta}$$

$$= \frac{\int \cdots \int \prod_{j \notin N} q_\phi(\theta_j)(1 - p(y_i|\theta_j)) \frac{1}{k} \sum_{l=1}^k \int \cdots \int p(y_i|\theta_{n_l}) \frac{\partial}{\partial \phi} q_\phi(\theta_{n_l}) \prod_{j=1}^{k, j \neq l} q_\phi(\theta_{n_j}) p(y_i|\theta_{n_j}) d\Theta_N d\Theta_{-N}}{\int \cdots \int \prod_{j \notin N} q_\phi(\theta_j)(1 - p(y_i|\theta_j)) \int \cdots \int \prod_{j=1}^k q_\phi(\theta_{n_j}) p(y_i|\theta_{n_j}) d\Theta_N d\Theta_{-N}}$$

$$= \frac{\int \cdots \int \prod_{j \notin N} q_\phi(\theta_j)(1 - p(y_i|\theta_j)) \frac{1}{k} \sum_{l=1}^k q_\phi^Y(y_i)^{k-1} \frac{\partial}{\partial \phi} q_\phi^Y(y_i) d\Theta_{-N}}{\int \cdots \int \prod_{j \notin N} q_\phi(\theta_j)(1 - p(y_i|\theta_j)) q_\phi^Y(y_i)^k d\Theta_{-N}}$$

$$= \frac{q_\phi^Y(y_i)^{k-1} \frac{\partial}{\partial \phi} q_\phi^Y(y_i)}{q_\phi^Y(y_i)^k}$$

$$= g_i^{\log}(\phi).$$

Collecting all cases of acceptance, we directly have $\mathbb{E}[g_i^{\text{RS}}] = g_i^{\log}(\phi)$. $\square$

At a high level, the gradient estimator in the main text is only asymptotically unbiased, while unnormalized rejection sampling is unbiased with any sample size, hence more suitable for SGD.

**Bounded likelihood.** The assumption on the bounded likelihood can be achieved at two general settings, (1) the observations are discrete thereby $p_i(y_i|\theta) \leq 1$, and (2) the likelihood has a fixed non-zero variance. For example in a one-dimensional linear regression problem, the likelihood is $p(y|\beta, \sigma) = \text{normal}(y|\beta X, \sigma)$, if $\sigma$ is fixed to be a small number, then $M = (\sqrt{2\pi}\sigma)^{-1}$.

A fixed $\sigma$ is less evil than it seems. First, the model is overparametrized, the observational noise can either be learned from $\sigma$, or the variational variance of the constant feature. Second, PVI allows a more general inference paradigm, where $\phi$ contains both the variational parameters and those parameters that we would wish to learn by point estimate, which will then contain this $\sigma$. Indeed, the rejection sampling idea is still applicable when $M = M(\phi)$ depends on $\phi$, and varies across SGD iterations.

In practice, however, we find that the benefits of having an unbiased estimator are not significant compared with the biased estimator in the main paper. This coincides with the findings of another unbiased estimator derived in Vandegar et al. (2021), which is also found not beneficial when the Monte Carlo sample size is large.

### A.3. Convergence of the logarithmic score gradient

For the logarithmic score

$$\mathcal{L}^{\log}(\phi) = \sum_{i=1}^{n} \log \int_{\Theta} p(y_i|\theta) q_\phi(\theta) d\theta,$$

we have a gradient estimator

$$g_M^{\log}(\phi) = \sum_{i=1}^{n} \frac{\sum_{j=1}^{M} \frac{d}{d\phi} p(y_i|T_\phi(u_j))}{\sum_{j=1}^{M} p(y_i|T_\phi(u_j))}$$

after reparameterization of $\theta = T_\phi(u)$ and with $M$ samples of $u_1, \ldots, u_M$. We first show that it converges in probability.

**Proposition 5.** *As $M \to \infty$, $g_M^{\log}(\phi) \xrightarrow{p} \frac{d}{d\phi} \mathcal{L}^{\log}(\phi)$.*

*Proof.* Define $\theta_j = T_\phi(u_j)$, $A_M = \frac{1}{M} \sum_{j=1}^{M} \frac{d}{d\phi} p(y|\theta_j)$, $B_M = \frac{1}{M} \sum_{j=1}^{M} p(y|\theta_j)$. To prove the convergence, it is enough to show for any $y$,

$$\frac{A_M}{B_M} \xrightarrow{p} \frac{d}{d\phi} \log \int_{\Theta} p(y|\theta) q_\phi(\theta) d\theta.$$

Note $A_M \xrightarrow{p} \int_{\Theta} \frac{d}{d\phi} p(y|\theta) q_\phi(\theta) d\theta$ and $B_M \xrightarrow{p} \int_{\Theta} p(y|\theta) q_\phi(\theta) d\theta$ under regularity conditions. By Slutsky's theorem, we have that $A_M / B_M \xrightarrow{p} \frac{d}{d\phi} \log \int_{\Theta} p(y|\theta) q_\phi(\theta) d\theta$. $\square$

Note the gradient estimator has the same form as a self-normalized importance sampling estimator, which converges almost surely. The bias of a self-normalized importance sampling estimator is also known to converge to zero approximately at the rate of $\mathcal{O}(M^{-1})$. See Owen (2013) for more details.

### A.4. Convergence of the quadratic score gradient

For the quadratic score

$$\mathcal{L}^{\text{quad}}(\phi) = \sum_{i=1}^{n} 2 \int_{\Theta} f(\theta, y_i) q_\phi(\theta) d\theta - \sum_{j=1}^{I} \left( \int_{\Theta} f(\theta, j) q_\phi(\theta) d\theta \right)^2,$$

we have a gradient estimator

$$g_M^{\text{quad}}(\phi) = 2 \sum_{i=1}^{n} \frac{1}{M} \sum_{j=1}^{M} \frac{d}{d\phi} f(T_\phi(u_j), y_i) - \sum_{j=1}^{I} \left( \frac{1}{M} \sum_{k=1}^{M} f(T_\phi(u_k), j) \right) \left( \frac{1}{M} \sum_{k=1}^{M} \frac{df(T_\phi(u_k), j)}{d\phi} \right).$$

We have the same result of converging in probability.

**Proposition 6.** *As $M \to \infty$, $g_M^{\text{quad}}(\phi) \xrightarrow{p} \frac{d}{d\phi}\mathcal{L}^{\text{quad}}(\phi)$.*

*Proof.* Define $\theta_k = T_\phi(u_k)$, $A_{M,j} = \frac{1}{M}\sum_{k=1}^{M}\frac{d}{d\phi}f(\theta_k, j)$, $B_{M,j} = \frac{1}{M}\sum_{k=1}^{M}f(\theta_k, j)$. Note for any $j$, $A_{M,j} \xrightarrow{p}$ $\frac{d}{d\phi}\int_\Theta f(\theta, j)q_\phi(\theta)d\theta$ and $B_{M,j} \xrightarrow{p} \int_\Theta f(\theta, j)q_\phi(\theta)d\theta$ under regularity conditions. Then, for a single $y$

$$\frac{d}{d\phi}\left(2\int_\Theta f(\theta, y)q_\phi(\theta)d\theta - \sum_{j=1}^{I}\left(\int_\Theta f(\theta, j)q_\phi(\theta)d\theta\right)^2\right)$$

$$= 2\frac{d}{d\phi}\int_\Theta f(\theta, y)q_\phi(\theta)d\theta - 2\sum_{j=1}^{I}\left(\int_\Theta f(\theta, j)q_\phi(\theta)d\theta\right)\frac{d}{d\phi}\left(\int_\Theta f(\theta, j)q_\phi(\theta)d\theta\right)$$

$$\approx 2A_{M,y} - 2\sum_{j=1}^{I}B_{M,j}A_{M,j}.$$

By Slutsky's theorem, we have that $2A_{M,y} - 2\sum_{j=1}^{I}B_{M,j}A_{M,j}$ converges in probability for a single $y$. Thus the constructed gradient estimator also converges in probability. $\qquad\square$

### A.5. Convergence of the interval score gradient

For the interval score objective

$$\mathcal{L}^{\text{IS}}(\phi) = -\sum_{i=1}^{n}(U_\alpha - L_\alpha) + \frac{2}{\alpha}(L_\alpha - y_i)\mathbb{I}(y_i < L_\alpha) + \frac{2}{\alpha}(y_i - U_\alpha)\mathbb{I}(y_i > U_\alpha), \tag{10}$$

the gradient estimator is

$$g_M^{\text{IS}}(\phi) = \sum_{i=1}^{n}\frac{\partial\hat{U}_\alpha}{\partial\phi}\left(\frac{2}{\alpha}\mathbb{I}(y_i > \hat{U}_\alpha) - 1\right) - \frac{\partial\hat{L}_\alpha}{\partial\phi}\left(\frac{2}{\alpha}\mathbb{I}(y_i < \hat{L}_\alpha) - 1\right), \tag{11}$$

where $\hat{L}_\alpha$ and $\hat{U}_\alpha$ are estimated from $M$ samples $y_1^{\text{sim}}, y_2^{\text{sim}}, ..., y_M^{\text{sim}}$ from $\int_\Theta p(y|\theta)q_\phi(\theta)d\theta$. We show that the gradient estimator is convergent.

**Proposition 7.** *As $M \to \infty$, $g_M^{\text{IS}}(\phi) \xrightarrow{p} \frac{d}{d\phi}\mathcal{L}^{\text{IS}}(\phi)$.*

*Proof.* There are two parts of the proof. First, $\frac{\partial\hat{L}_\alpha}{\partial\phi}$ and $\frac{\partial\hat{U}_\alpha}{\partial\phi}$ are obtained from autodiff. We show they are consistent. Consider the empirical quantile function $\hat{Q}_\gamma(\phi)$ for the $M$ simulated samples. For any $\phi$, by Glivenko-Cantelli theorem, $\hat{Q}_\gamma(\phi) \to Q_\gamma(\phi)$ almost surely. Under regularity conditions, with Attouch theorem, $\partial\hat{Q}_\gamma(\phi)$ converges graphically to $\partial Q_\gamma(\phi)$, which has a unique element $\frac{\partial Q_\gamma(\phi)}{\partial\phi}$. By applying this with $\gamma = \alpha/2$ and $\gamma = 1 - \alpha/2$, we have both $\frac{\partial\hat{L}_\alpha}{\partial\phi}$ and $\frac{\partial\hat{U}_\alpha}{\partial\phi}$ are convergent.

Next, $\mathbb{I}(y_i > \hat{U}_\alpha)$ and $\mathbb{I}(y_i < \hat{L}_\alpha)$ converge in probability to the true indicator functions $\mathbb{I}(y_i > U_\alpha)$ and $\mathbb{I}(y_i < L_\alpha)$. So the combination of convergent gradient estimators converges in probability. This concludes the proof. $\qquad\square$

### A.6. Unbiasedness of the CRPS gradient

For the CRPS objective

$$\mathcal{L}^{\text{CRPS}}(\phi) = -\sum_{i=1}^{n}\mathbb{E}_{y^{\text{sim}}\sim\int_\Theta p(y|\theta)q_\phi(\theta)d\theta}\left[|y^{\text{sim}} - y_i|\right] + \frac{1}{2}\mathbb{E}_{y_1^{\text{sim}}, y_2^{\text{sim}}\sim\int_\Theta p(y|\theta)q_\phi(\theta)d\theta}\left[|y_1^{\text{sim}} - y_2^{\text{sim}}|\right],$$

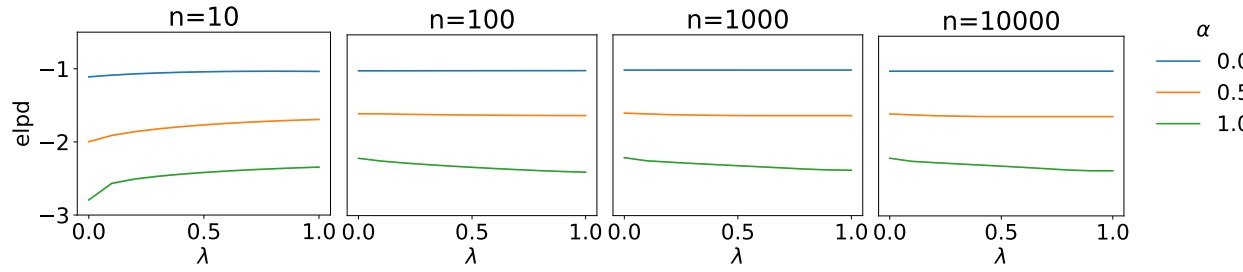

*Figure 7.* Test average elpd on a simple regression model, with different dataset sizes $n \in \{10, 100, 1000, 10000\}$ and specification scales $\alpha$ ($\alpha = 0$ for well-specified and $\alpha = 1$ for misspecified), interpolating between PVI ($\lambda = 0$) and VI ($\lambda = 1$). It is found that VI gives a better finite sample performance but PVI works better asymptotically.

we have an unbiased estimator, assuming $y^{\text{sim}}$ can be generated with reparameterization of $y^{\text{sim}} = \mathcal{T}_\phi(\epsilon)$, $\epsilon \sim q(\cdot)$. Such reparameterization is performed at both the variational distribution $q_\phi$ and the simulator $p(y|\theta)$. Then for an arbitrary $y$,

$$\frac{d}{d\phi} \mathbb{E}_{y^{\text{sim}} \sim \int_\Theta p(y|\theta)q_\phi(\theta)d\theta} \left[ |y^{\text{sim}} - y_i| \right] = \mathbb{E}_{\epsilon \sim q(\cdot)} \left[ \frac{d}{d\phi} |\mathcal{T}_\phi(\epsilon) - y| \right]$$

$$\approx \frac{1}{2M} \sum_{m=1}^{2M} \text{sign}(\mathcal{T}_\phi(\epsilon_m) - y) \frac{d}{d\phi} \mathcal{T}_\phi(\epsilon_m)$$

$$= \frac{1}{2M} \sum_{m=1}^{2M} \text{sign}(y_m^{\text{sim}} - y) \frac{d}{d\phi} y_m^{\text{sim}}.$$

Similarly, we have

$$\frac{d}{d\phi} \mathbb{E}_{y_1^{\text{sim}}, y_2^{\text{sim}} \sim \int_\Theta p(y|\theta)q_\phi(\theta)d\theta} \left[ |y_1^{\text{sim}} - y_2^{\text{sim}}| \right] = \mathbb{E}_{\epsilon_a, \epsilon_b \sim q(\cdot)} \left[ \frac{d}{d\phi} |\mathcal{T}_\phi(\epsilon_a) - \mathcal{T}_\phi(\epsilon_b)| \right]$$

$$\approx \frac{1}{M} \sum_{m=1}^{M} \text{sign}(\mathcal{T}_\phi(\epsilon_m) - \mathcal{T}_\phi(\epsilon_{m+M})) \frac{d}{d\phi} (\mathcal{T}_\phi(\epsilon_m) - \mathcal{T}_\phi(\epsilon_{m+M}))$$

$$= \frac{1}{M} \sum_{m=1}^{M} \text{sign}(y_m^{\text{sim}} - y_{m+M}^{\text{sim}}) \frac{d}{d\phi} (y_m^{\text{sim}} - y_{m+M}^{\text{sim}}).$$

Both Monte Carlo estimators are unbiased and we directly have an unbiased gradient estimator for CRPS:

$$g^{\text{CRPS}}(\phi) = -\frac{1}{2M} \sum_{i=1}^{n} \sum_{m=1}^{2M} \left( g_m \cdot \text{sign}(y_m^{\text{sim}} - y_i) \right) + \frac{1}{2M} \sum_{m=1}^{M} \left( (g_m - g_{m+M}) \cdot \text{sign}(y_m^{\text{sim}} - y_{m+M}^{\text{sim}}) \right),$$

where $g_m = \frac{d}{d\phi} y_m^{\text{sim}}$.

# B. Additional Results and Experiments

We explore further properties of PVI through three simulated regression models, an example in linear regression, and additional benchmarks.

## B.1. Interpolation between VI and PVI

Posterior regularization allows us to interpolate between VI and PVI. We expect that VI has good finite sample performance because it incorporates the prior while PVI has better performance asymptotically when the model is misspecified. We consider the regression problem:

$$y_i = \mathbf{x}_i^T \boldsymbol{\beta} + \epsilon, \ \epsilon \sim \text{Normal}(0, 1),$$

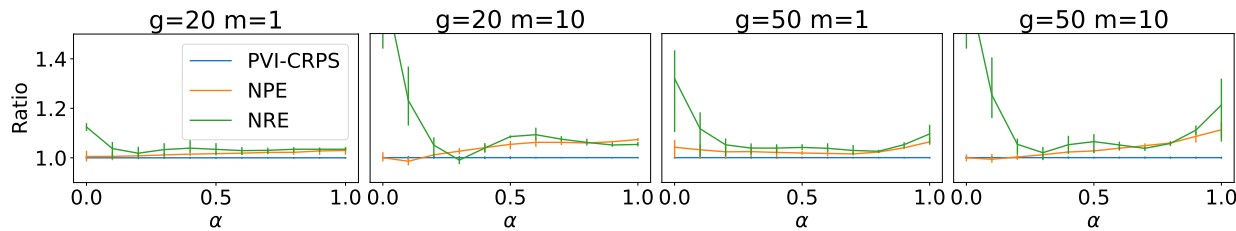

*Figure 8.* Ratios of test CRPS with respect to the test CRPS of PVI, with different parameters of the $y^2$ regression model (group number $g$, observation dimension $m$, misspecification scale $\alpha$). We compare against NPE and NRE from SBI, showing that PVI is more favored especially when the model is misspecified ($\alpha \to 1$).

where $y_i \in \mathbb{R}$, $\mathbf{x}_i \in \mathbb{R}^d$, $\boldsymbol{\beta} \in \mathbb{R}^d$ for datapoint $i \in \{1, 2, ..., n\}$. In the data generating process, we specify a parameter $\alpha$ to interpolate between a well-specified model ($\alpha = 0$) and a misspecified model ($\alpha = 1$). We generate a collection of parameters $\{\boldsymbol{\beta}_0, \boldsymbol{\beta}_1, ..., \boldsymbol{\beta}_g\}$ and choose $\boldsymbol{\beta}$ from $\{(1 - \alpha)\boldsymbol{\beta}_0 + \alpha\boldsymbol{\beta}_j \mid j = 1, 2, ..., g\}$ when generating each datapoint. The parameters are learned by a combined loss of VI and PVI. Namely, we maximize

$$\mathcal{L}_\lambda(\phi) = \lambda\mathcal{L}_{\mathrm{VI}}(\phi) + (1 - \lambda)\mathcal{L}_{\mathrm{PVI}}(\phi).$$

In Figure 7, the average elpd on the test set is plotted under different settings. We first focus on the dataset size. When $n = 10$, we note that the full PVI gives worse test predictive scores. This is reasonable as the learned parameters from PVI are entirely determined by limited number of samples, while VI is regularized by the prior. As the sample size $n$ increases, PVI outperforms VI since it directly targets the prediction. We also note that on the correctly specified model $\alpha = 0$, PVI and VI are asymptotically equivalent. In practice, it is reasonable to have a combined objective of PVI and VI (which we call PVI with posterior regularization) for robust performance under different sample sizes and different models.

### B.2. Likelihood-free regression

In regression problems, sometimes we are only able to access the summary statistics of groups of outcomes. This scenario is common in differential privacy and meta-analysis. We consider the same regression problem with more dimensions in the observation:

$$\mathbf{y}_i = \mathbf{X}_i\boldsymbol{\beta} + \boldsymbol{\epsilon}, \ \boldsymbol{\epsilon} \sim \mathrm{MVN}(\mathbf{0}, \mathbf{I}),$$

where $\mathbf{y}_i \in \mathbb{R}^m$, $\mathbf{X}_i \in \mathbb{R}^{m \times d}$, $\boldsymbol{\beta} \in \mathbb{R}^d$ for datapoint $i \in \{1, 2, ..., n\}$. However, instead of the full vector $\mathbf{y}_i$, we only have access to $\widehat{y_i^2} := \sum_{j=1}^m y_{i,j}^2$. The likelihood of $\widehat{y_i^2}$ follows non-central chi-squared distribution, whose probability density function involves an infinite summation. What's worse, the model may be misspecified. In the data generating process, we specify a parameter $\alpha$ to interpolate between a well-specified model ($\alpha = 0$) and a misspecified model ($\alpha = 1$). We generate a collection of parameters $\{\boldsymbol{\beta}_0, \boldsymbol{\beta}_1, ..., \boldsymbol{\beta}_g\}$ and choose $\boldsymbol{\beta}$ from $\{(1 - \alpha)\boldsymbol{\beta}_0 + \alpha\boldsymbol{\beta}_j \mid j = 1, 2, ..., g\}$ for each datapoint. We compare PVI with CRPS against the standard implementations of neural posterior estimation (NPE) (Papamakarios & Murray, 2016) and neural ratio estimation (NRE) (Hermans et al., 2020) from SBI (Tejero-Cantero et al., 2020). Due to the Bayesian nature and the misspecification of model their sequential counterparts perform worse.

First, we compare the performances of different methods with different interpolation parameters $\alpha$. We find that PVI with CRPS has a robust performance under different levels of misspecification. In addition, the performance gain is larger with more dimensions in observations (higher $m$). Note that the performance of NRE is good only with a modest level of misspecification. The good performance of NRE under modestly misspecified case aligns with the findings in Cannon et al. (2022), but we also show that eventually NRE can be as bad as NPE, as the problem of misspecification worsens.

### B.3. Properties of different scoring rules

In theory, as $n \to \infty$, PVI will converge to the optimal parameter regardless of the scoring rule. In practice, different scoring rules have different finite sample behaviors and properties. We demonstrate the differences with a simple regression model. Suppose we have a true model $\mathbf{y}_i = \mathbf{x}_i^T\boldsymbol{\beta} + \epsilon$ where $\epsilon \sim t_{\mathrm{df}}$. The model is misspecified in two senses: there are multiple $\boldsymbol{\beta}$s to generate data, and we use a normal model during inference. PVI is capable of handling these challenges. We find that the log score leads to the best test log likelihood. More interestingly, different scores lead to different test coverages. When

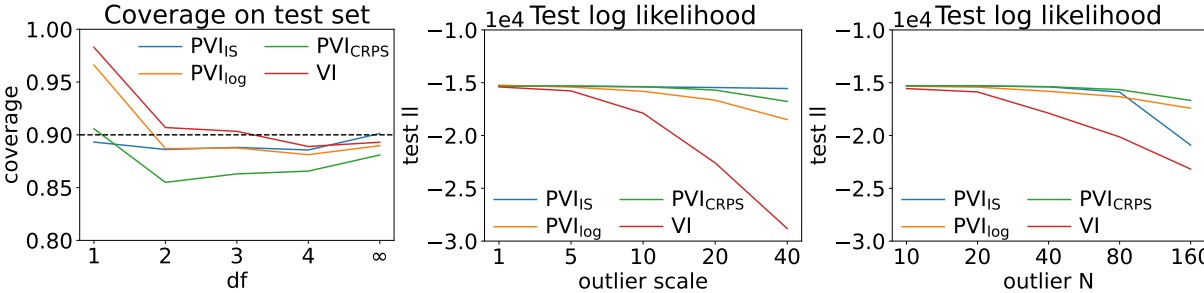

*Figure 9.* Comparing different scores with a simulated regression model. Left: test coverage from PVI with different objectives with different Student-t noise models. Middle: test log likelihood from PVI with different objectives after mixing training data with 40 outliers of different scales. Right: test log likelihood from PVI with different objectives after mixing training data with different numbers of outliers of scale 10. In these experiments $n = 1000$. IS is trained with the significance level of $\alpha = 0.1$.

training with IS at the confidence level of $\alpha = 0.1$, we consistently get correct test coverage that is not achieved by the other scoring rules. See the first panel of Figure 9. This result demonstrates that PVI can potentially provide free conformal prediction with proper settings of the scoring rule. Next, we study the robustness of different scoring rules. During the data generating process, we insert noisy data of different scales into the training set. This mimics the real-world challenge of working with contaminated data. We first insert $4\%$ of outliers into the training set and vary the scale of outliers. The test log likelihoods of different methods are in the second panel of Figure 9. IS at the significance level of $\alpha = 0.1$ is the most robust to this type of contamination, followed by CRPS. This is because both IS and CRPS care about the calibration of the predictive distribution. Next, we keep the scale of outliers and vary the number of outliers. When there are $16\%$ contaminated training data, the performance of IS with $\alpha = 0.1$ becomes much worse, as shown in Figure 9. CRPS is found to be the most robust to different sizes of contaminated data. This is because IS only works for a single significance level, while CRPS focuses on the global calibration at different locations.

## B.4. The expressiveness of PVI

Another interesting view is that PVI expands the expressiveness of the underlying models by implicitly adding a layer of trainable parameters. We demonstrate this with an example in linear regression. Suppose our model is

$$y|x \sim \text{normal}(wx + b, \sigma),$$

and we have the variational distribution $q_\phi(\theta)$ where $\theta = [w, b, \sigma^2]$. If we assign a distribution over $b$ and $\sigma$, the noise distribution is expanded to be a continuous mixture of normal distributions. PVI subsumes this case, which means if the posterior distribution is flexible enough, PVI can model arbitrary noise distributions for linear regression. This property can be generalized to other models with separable noise. Moreover, PVI also defines a distribution over $w$. In the general case, we have

$$\mathbb{E}_{\int p(y|x,\theta)q_\phi(\theta)d\theta}[y] = \int \mathbb{E}_{p(y|x,\theta)}[y]q_\phi(\theta)d\theta$$
$$= \mathbb{E}_{q_\phi(\theta)}[w]x + \mathbb{E}_{q_\phi(\theta)}[b],$$
$$\text{Var}_{\int p(y|x,\theta)q_\phi(\theta)d\theta}[y] = \text{Var}_{q_\phi(\theta)}[\mathbb{E}_{p(y|x,\theta)}[y]] + \mathbb{E}_{q_\phi(\theta)}[\text{Var}_{p(y|x,\theta)}[y]]$$
$$= \text{Var}_{q_\phi(\theta)}[wx + b] + \mathbb{E}_{q_\phi(\theta)}[\sigma^2]$$
$$= \text{Var}_{q_\phi(\theta)}[w]x^2 + 2\text{Cov}_{q_\phi(\theta)}(w, b)x + \text{Var}_{q_\phi(\theta)}[b] + \mathbb{E}_{q_\phi(\sigma)}[\sigma^2].$$

Although the resulting model still has a linear mean, the noise distribution can vary by location $x$, which is more expressive than models with stationary noise.

## B.5. Benchmarking Stan models

We test PVI against vanilla VI on a collection of models from `posteriordb` (Magnusson et al., 2024) and evaluate their held-out data prediction performance. For each model, we use the same $M = 100$ Monte Carlo batch size. We perform mild hyperparameter tuning for PVI (regularizer $r \in \{r^{\text{prior}}, r^{\text{post}}\}$, $\lambda \in \{0, 0.01, 0.1, 1\}$) by cross-validation. We consider

| Model | Method | Diagonal covariance | | | | Dense covariance | | | |
|---|---|---|---|---|---|---|---|---|---|
| | | Log (↑) | Quad (↑) | CRPS (↓) | IS (×10³) (↓) | Log (↑) | Quad (↑) | CRPS (↓) | IS (×10³) (↓) |
| election | VI | -1494.44 (0.01) | 0.21 (0.00) | - | - | -1493.80 (0.16) | 0.21 (0.00) | - | - |
| | PVI-Log | **-1490.89** (0.01) | 0.21 (0.00) | - | - | **-1493.64** (0.09) | 0.20 (0.00) | - | - |
| | PVI-Quad | -1603.60 (1.96) | **0.25** (0.00) | - | - | -1603.64 (2.04) | **0.25** (0.00) | - | - |
| glmm | VI | -1159.49 (0.60) | - | - | - | -1161.63 (0.81) | - | - | - |
| | PVI-Log | **-1150.21** (0.09) | - | - | - | **-1149.00** (0.58) | - | - | - |
| earnings | VI | -298.12 (0.05) | - | 111.51 (0.03) | 229.89 (0.12) | -298.18 (0.07) | - | 111.60 (0.04) | 229.94 (0.09) |
| | PVI-Log | **-288.69** (0.01) | - | **108.49** (0.00) | 233.88 (0.08) | **-283.42** (0.60) | - | 109.08 (0.17) | 232.43 (0.53) |
| | PVI-CRPS | -291.54 (0.12) | - | 109.55 (0.03) | 232.72 (0.18) | -284.29 (0.29) | - | **108.78** (0.07) | 231.98 (0.61) |
| | PVI-IS | -1046.23 (897.24) | - | 125.89 (0.60) | **228.13** (3.64) | -310.03 (2.82) | - | 121.44 (1.76) | **229.32** (4.41) |
| kidscore | VI | -590.67 (0.08) | - | 4481.15 (0.59) | 3664.73 (1.36) | -590.71 (0.07) | - | 4480.55 (0.50) | 3664.71 (1.11) |
| | PVI-Log | **-369.91** (0.04) | - | **872.82** (0.60) | 778.14 (0.75) | -371.06 (0.11) | - | 871.84 (0.93) | 771.42 (1.76) |
| | PVI-CRPS | -370.27 (0.05) | - | 875.74 (0.56) | 752.10 (1.33) | **-370.51** (0.16) | - | **867.66** (1.01) | 764.86 (1.79) |
| | PVI-IS | -374.63 (0.05) | - | 925.96 (0.64) | **717.11** (0.43) | -382.05 (0.14) | - | 1001.33 (1.05) | **722.39** (0.30) |
| nes | VI | -185.71 (0.01) | - | 91.42 (0.01) | 94.83 (0.03) | -185.71 (0.03) | - | 91.49 (0.02) | 94.79 (0.15) |
| | PVI-Log | **-185.60** (0.09) | - | 91.02 (0.01) | 96.90 (0.82) | **-184.87** (0.02) | - | 90.70 (0.08) | 96.98 (0.16) |
| | PVI-CRPS | -187.15 (0.76) | - | **87.81** (0.09) | 102.81 (1.05) | -187.91 (0.66) | - | **89.75** (0.53) | 96.21 (2.75) |
| | PVI-IS | -226.56 (22.08) | - | 110.31 (0.25) | **65.37** (0.70) | -194.99 (0.16) | - | 104.42 (0.26) | **68.41** (0.31) |
| radon | VI | -3412.90 (0.11) | - | 1338.01 (0.09) | **35635.36** (40.07) | **-3411.69** (0.09) | - | 1337.71 (0.05) | **36044.18** (30.63) |
| | PVI-Log | **-3372.10** (0.08) | - | 1333.02 (0.03) | 41141.75 (34.70) | -3415.79 (0.21) | - | 1344.73 (0.16) | 37498.42 (111.74) |
| | PVI-CRPS | -3482.39 (2.41) | - | **1323.71** (0.30) | 39140.21 (45.48) | -3495.78 (18.63) | - | **1330.10** (5.81) | 39010.02 (124.78) |
| | PVI-IS | -3444.50 (0.49) | - | 1363.77 (0.09) | 37070.83 (57.80) | -3487.35 (1.21) | - | 1384.58 (0.63) | 37367.99 (32.33) |
| wells | VI | -393.64 (0.04) | 0.18 (0.00) | - | - | -393.68 (0.01) | 0.18 (0.00) | - | - |
| | PVI-Log | **-392.14** (0.02) | 0.18 (0.00) | - | - | **-391.02** (0.02) | 0.18 (0.00) | - | - |
| | PVI-Quad | -415.39 (0.77) | **0.27** (0.01) | - | - | -415.40 (0.77) | **0.27** (0.01) | - | - |

*Table 2.* Test predictive score (mean and std from 5 random seeds) on 7 models from posteriordb with VI or PVI. We show that PVI is better in general when the prediction is evaluated in the corresponding scoring rule.

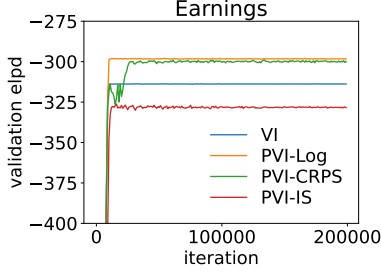

*Figure 10.* Validation log predictive density as a function of training iteration for the `earnings` model.

| Method | Training time (min) |
|---|---|
| VI | 1.96 (0.04) |
| PVI-Log | 4.11 (0.07) |
| PVI-CRPS | 10.40 (0.31) |
| PVI-IS | 57.08 (0.39) |

*Table 5.* Training time of different methods for the `earnings` model for 200,000 iterations.

two variational families, Gaussian with diagonal and dense covariance matrices, and compare each of the test scores on held-out data when possible. The results of both cases are shown in Table 2. We find that in general PVI always improves the prediction performance of the corresponding score on held-out data, a free model improvement. There are cases where PVI and VI are close (e.g., the election model and the radon model), which indicates that the model is nearly well-specified. Indeed, the election model here is saturated and passes the posterior predictive check as well.

We studied the effect of regularization in this experiment. Across the models, we did not find a clear pattern for whether the prior or posterior regularization works better (Table 3). We view this extra binary tuning as added flexibility. For completeness, we also add results using Hamiltonian Monte Carlo (NUTS) with the exact posterior as another alternative. This additional experiment is shown as Table 4. We do not see a qualitative jump in performance from VI to NUTS.

### B.6. Additional demonstration of the `earnings` model

We use the `earnings` model to demonstrate additional properties of our method. First, in terms of per iteration time during training, PVI with log score is comparable to vanilla VI. However, PVI with CRPS or IS is slower. To illustrate this empirically, we include additional results showing convergence behavior and training time under the log score, CRPS, and IS (Figure 10 and Table 5). Second, we empirically observe that the posteriors from different scoring rules usually overlap with each other but are not exactly the same. At a high level, this solution diversity reflects the benefit of multiple scoring rules. To be concrete, in Figure 11, we compare PVI solutions in the model under various scoring rules. While most

| Model | Method | Diagonal covariance | | Dense covariance | |
|---|---|---|---|---|---|
| | | $r$ | $\lambda$ | $r$ | $\lambda$ |
| election | PVI-Log | $r^{\text{post}}$ | 0.01 | $r^{\text{post}}$ | 0.01 |
| | PVI-Quad | $r^{\text{prior}}$ | 1.0 | $r^{\text{prior}}$ | 1.0 |
| glmm | PVI-Log | $r^{\text{post}}$ | 0.1 | $r^{\text{post}}$ | 0.1 |
| earnings | PVI-Log | $r^{\text{prior}}$ | 0.01 | $r^{\text{prior}}$ | 1.0 |
| | PVI-CRPS | $r^{\text{post}}$ | 0.01 | $r^{\text{prior}}$ | 0.1 |
| | PVI-IS | $r^{\text{prior}}$ | 0.01 | $r^{\text{post}}$ | 0.1 |
| kidscore | PVI-Log | $r^{\text{post}}$ | 0.01 | $r^{\text{post}}$ | 0.01 |
| | PVI-CRPS | $r^{\text{post}}$ | 0.1 | $r^{\text{post}}$ | 0.1 |
| | PVI-IS | $r^{\text{post}}$ | 1.0 | $r^{\text{post}}$ | 1.0 |
| nes | PVI-Log | $r^{\text{post}}$ | 0.01 | $r^{\text{post}}$ | 0.1 |
| | PVI-CRPS | $r^{\text{post}}$ | 0.1 | $r^{\text{post}}$ | 0.01 |
| | PVI-IS | $r^{\text{post}}$ | 0.1 | $r^{\text{post}}$ | 1.0 |
| radon | PVI-Log | $r^{\text{prior}}$ | 1.0 | $r^{\text{post}}$ | 1.0 |
| | PVI-CRPS | $r^{\text{prior}}$ | 1.0 | $r^{\text{prior}}$ | 1.0 |
| | PVI-IS | $r^{\text{post}}$ | 1.0 | $r^{\text{post}}$ | 0.1 |
| wells | PVI-Log | $r^{\text{post}}$ | 0.01 | $r^{\text{prior}}$ | 0.1 |
| | PVI-Quad | $r^{\text{post}}$ | 0.0 | $r^{\text{post}}$ | 0.0 |

*Table 3.* Optimal regularization setting for PVI on the 7 models from posteriordb.

| Model | Method | Log ($\uparrow$) | Quad ($\uparrow$) | CRPS ($\downarrow$) | IS($\downarrow$) |
|---|---|---|---|---|---|
| election | NUTS | -1494.89 (1.35) | 0.21 (0.00) | - | - |
| glmm | NUTS | -1195.54 (2.22) | - | - | - |
| earnings | NUTS | -298.03 (0.02) | - | 111.34 (0.06) | 229.68 (0.06) |
| kidscore | NUTS | -591.16 (0.06) | - | 4482.96 (2.31) | 3638.49 (5.87) |
| nes | NUTS | -185.69 (0.01) | - | 91.75 (0.11) | 94.41 (0.09) |
| radon | NUTS | -3445.43 (69.26) | - | 1347.92 (27.97) | 35905.94 (427.25) |
| wells | NUTS | -393.77 (0.01) | 0.18 (0.00) | - | - |

*Table 4.* Test predictive score (mean and std from 5 random seeds) on 7 models from posteriordb with NUTS using the default hyperparameters in NumPyro.

posteriors have similar means (e.g., $\beta_0$ is close to 9), the posterior uncertainty differs across scoring rules, each serving a different need in prediction.

### B.7. Scalability of predictive variational inference

PVI inherits the scalability properties of the underlying variational family. In particular, using simpler families (e.g., mean-field Gaussian), it can scale to higher dimensions. As a check, we implemented log-score PVI in a normal model with parameter dimension up to 10,000 (see Figure 12). We considered a model $p(y \mid \theta) = \text{MVN}(0, I_d)$ with IID data $y_i \sim \text{MVN}(0, 2\ I_d)$ and used a mean-field Gaussian variational family ($n = 100$ and $M = 10$). As the dimension of $\theta$ increases, the expected log predictive density (elpd; higher is better) scales linearly for both VI and PVI. Across all dimensions, PVI consistently achieves higher elpd than VI, indicating improved predictive performance. While this experiment is simple, it demonstrates that PVI can be applied in high-dimensional settings. It remains an important direction for future work to practically push the scalability of PVI in modern deep models.

## C. Experiment settings

We detail the training settings of our major experiments. Since the experiments involve stochastic optimization and possibly mini-batching, we use the averaged score (or the empirical score) instead of the summations of scores as objectives in

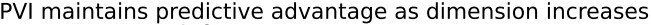

*Figure 11.* Posterior distribution of parameters for the `earnings` model with different methods using mean-field Gaussian variational family.

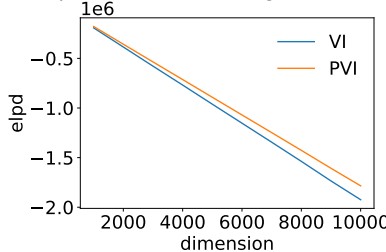

*Figure 12.* We compare log-score PVI and vanilla VI across varying parameter dimensions (up to 10,000). We consider the model $p(y \mid \theta) = \text{MVN}(0, I_d)$ with IID data $y_i \sim \text{MVN}(0, 2\, I_d)$, using a mean-field variational family ($n = 100$ and $M = 10$). As we increase the dimension of $\theta$, the expected log predictive density (elpd, higher is better) of the induced predictive distributions scales linearly for both VI and PVI. Across all dimensions, PVI consistently achieves higher elpd than VI, indicating improved predictive performance.

practice. The training involving the regression models and the posteriordb models are performed on a single Intel(R) Xeon(R) Gold 6126 CPU. The training involving the cryoEM model is performed on a single NVIDIA Tesla A100 GPU.

### C.1. Election models

For the election models in Section 4.1, we train VI and PVI for $200,000$ iterations using the learning rate of $10^{-5}$, the RMSProp optimizer (Hinton et al., 2012) and Monte Carlo sample size $M = 100$. After $100,000$ iterations the learning rate is reduced to $10^{-6}$. The posterior is chosen to be full-rank Gaussian and no regularization is used for PVI.

### C.2. cryoEM models

Each image of the generated protein is a gray-scale image with shape 128x128. To simulate real images, a defocus parameter of the microscope is sampled from $[1000, 2000]$ and Gaussian noise is added to the rendered image to reduce the SNR. See Figure 13 for generated noisy images under different noise scales. The noise in the images makes the inference task very challenging.

Figure 1 in the main paper demonstrates the concentration of the posterior from the model $p^{\text{prior}}(\theta) \prod_{i=1}^{n} p(y_i | \theta)$. Because the likelihood is assumed unavailable in this model, the Bayes posterior is approximated by learning a ResNet (He et al.,

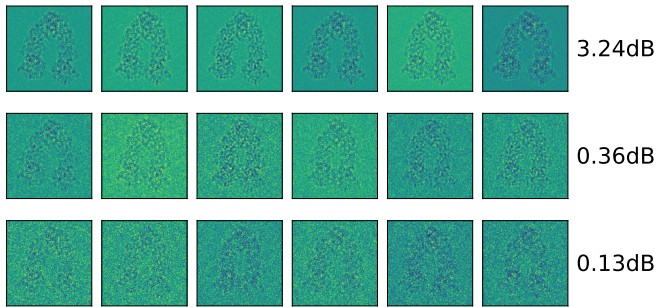

*Figure 13.* Examples of generated HSP90 protein images under different SNRs.

| Model | election | glmm | earnings | kidscore | nes | radon | wells |
|---|---|---|---|---|---|---|---|
| Likelihood | Bernoulli | Poisson | Normal | Normal | Normal | Normal | Bernoulli |
| $n$ | 11566 | 2072 | 1192 | 434 | 476 | 12573 | 3020 |
| Initial learning rate | $10^{-5}$ | $10^{-5}$ | $10^{-4}$ | $10^{-4}$ | $10^{-5}$ | $10^{-5}$ | $10^{-5}$ |

*Table 6.* Specifications of the posteriordb models.

2016) based neural ratio estimation (Hermans et al., 2020) with a convolutional embedding network and sampling using 10 independent chains of the No-U-Turn sampler (NUTS) (Hoffman & Gelman, 2014).

In all of the cryoEM experiments with the CRPS, we use Monte Carlo sample size $M = 32$ and mini-batch size 32 for the training objective. The variational distribution is chosen to be one-dimensional neural spline (Durkan et al., 2019) with 32 knots. The optimizer is Adam (Kingma & Ba, 2015) and the learning rate is scheduled with a linear warm-up to $10^{-3}$ in the first 1,000 steps, and a cosine annealing reducing to $10^{-4}$ in the remaining 9,000 steps.

### C.3. Posteriordb models

For each of the datasets, we split 60/20/20 as training, validation and test sets. VI and PVI are trained on the posteriordb models for 200, 000 iterations using the RMSProp optimizer and Monte Carlo sample size $M = 100$. Each model has a different learning rate as in Table 6. After 100, 000 iterations the learning rate is reduced to 0.1 of the starting learning rate.

### C.4. Additional regression models

In Supplement B.1, we demonstrate the interpolation between VI and PVI. Each of the experiments is performed with a learning rate of $10^{-3}$ for 20,000 iterations using the RMSProp optimizer.

In Supplement B.2, we compare PVI with CRPS against various baselines on a likelihood-free regression model. The dimension of the parameter $\boldsymbol{\beta}$ is chosen to be $d = 5$ and the number of observations is $n = 1, 000$. The distribution of the parameter is approximated by a neural spline flow (Durkan et al., 2019). PVI is trained with $M = 100$ Monte Carlo samples using the Adam optimizer (Kingma & Ba, 2015) and the learning rate is scheduled with a linear warm-up to $10^{-5}$ in the first 1,000 steps, and a cosine annealing reducing to $10^{-6}$ in the remaining 49,000 steps. To stabilize the training of the flow, the global norm of the gradient is clipped to be at most 10. For the baselines, we use 50,000 simulated samples to train NPE, and 10,000 simulated samples to train NRE. The numbers are chosen to match the training and inference time of PVI. In addition, we use NUTS to sample from the model implied by NRE.

## D. Formulations of the posteriordb models

We include the mathematical formulas of the posteriordb models in this section. The prior, if not specified, is chosen to be normal$(0, 1)$ for each variable.

### D.1. `election`

The election model uses data from seven CBS News polls conducted during the week before the 1988 U.S. presidential election (Gelman, 2007). There are in total $n = 11566$ datapoints. For data $i$, the age group $a_i \in \{1, 2, 3, 4\}$, education level $e_i \in \{1, 2, 3, 4\}$, state $s_i \in \{1, \ldots, 51\}$, region $r_i \in \{1, 2, 3, 4, 5\}$, indicator of black ethnicity $b_i \in \{0, 1\}$, gender $g_i \in \{0, 1\}$, the Republican share of the vote for president in the state in the previous election $p_i \in [0, 1]$, and voter outcome $y_i \in \{0, 1\}$ are collected. The likelihood model is

$$y_i \sim \text{Bernoulli}(\text{Logit}^{-1}(\beta_0 + \beta_1 b_i + \beta_2 g_i + \beta_3 b_i g_i + \beta_4 p_i + \beta_{5,a_i} + \beta_{6,e_i} + \beta_{7,a_i,e_i} + \beta_{8,s_i} + \beta_{9,r_i})).$$

For $\beta_0, \beta_1, \beta_2, \beta_3, \beta_4$, we assign the prior normal$(0, 100)$. For $\beta_{5,\cdot}$, we assign the prior normal$(0, \sigma_a)$ where $\sigma_a \sim$ log_normal$(3, 1)$. Similarly, for $\beta_{6,\cdot}$, we assign the prior normal$(0, \sigma_e)$ where $\sigma_e \sim$ log_normal$(3, 1)$. For $\beta_{7,\cdot,\cdot}$, we assign the prior normal$(0, \sigma_{ac})$ where $\sigma_{ac} \sim$ log_normal$(3, 1)$. For $\beta_{8,\cdot}$, we assign the prior normal$(0, \sigma_s)$ where $\sigma_s \sim$ log_normal$(3, 1)$. For $\beta_{9,\cdot}$, we assign the prior normal$(0, \sigma_r)$ where $\sigma_r \sim$ log_normal$(3, 1)$.

### D.2. `glmm`

The dataset contains simulated population counts of peregrines in various sites and years in the French Jura over 9 years (Kéry & Schaub, 2011). There are in total $n = 2072$ observations. For observation $i$, the site $s_i \in \{1, \ldots, 235\}$, the year $y_i \in \{1, \ldots, 9\}$ and the observed count $c_i$ are generated. The likelihood model is

$$c_i \sim \text{Poisson}(\exp(\mu + \alpha_{s_i} + \epsilon_{y_i})).$$

We assign $\text{normal}(0, 10)$ as the prior for $\mu$, $\text{normal}(0, \sigma_s)$ as the prior for $\alpha_\cdot$, $\text{normal}(0, \sigma_y)$ as the prior for $\epsilon_\cdot$, $\log\_\text{normal}(1, 1)$ as the prior for $\sigma_s$, and $\log\_\text{normal}(0, 1)$ as the prior for $\sigma_y$.

### D.3. `earnings`

The dataset studies the effects of heights and gender on income in dollars (Gelman, 2007). There are in total $n = 1192$ observations. Each observation $i$ contains the height $h_i \in \mathbb{R}^+$, the gender $g_i \in \{0, 1\}$ and the income $e_i \in \mathbb{R}^+$. The model is

$$\log e_i \sim \text{normal}(\beta_0 + \beta_1 h_i + \beta_2 g_i + \beta_3 h_i g_i, \sigma).$$

During processing of the dataset the heights are centered, so we assign $\text{normal}(0, 1)$ to all coefficients and $\log\_\text{normal}(0, 1)$ to $\sigma$ as priors.

### D.4. `kidscore`

The dataset contains cognitive test scores of three and four-year-old children (Gelman, 2007). Data is collected from $n = 434$ kids. For kid $i$, the IQ $q_i \in \mathbb{R}^+$, mother's IQ $m_i \in \mathbb{R}^+$ and mother's completion of high school $h_i \in \{0, 1\}i$ are collected. The likelihood model is

$$q_i \sim \text{normal}(\beta_0 + \beta_1 h_i + \beta_2 m_i + \beta_3 h_i m_i, \sigma).$$

Also, we centered the data so $\text{normal}(0, 1)$ is assigned as prior to all coefficients and $\text{normal}^+(0, 1)$ is assigned to $\sigma$ as prior.

### D.5. `nes`

The dataset contains the party identification information of individuals from the National Election Study of the United States in the year 2000 (Gelman, 2007). There are $n = 476$ datapoints. Each datapoint $i$ includes the party identification $p_i \in \{1, \ldots, 7\}$ (1 is strong Democrat and 7 is strong Republican), real political ideology $d_i \in \{1, \ldots, 7\}$ (1 is strong liberal and 7 is strong conservative), race $r_i \in \{0, 0.5, 1\}$, education level $e_i \in \{1, 2, 3, 4\}$, gender $g_i \in \{0, 1\}$, income percentile $l_i \in \{1, 2, 3, 4, 5\}$, age group $a_i \in \{1, 2, 3, 4\}$. The likelihood model contains multiple terms

$$p_i \sim \text{normal}(\beta_0 + \beta_1 d_i + \beta_2 r_i + \beta_3 \mathbb{I}(a_i = 2) + \beta_4 \mathbb{I}(a_i = 3) + \beta_5 \mathbb{I}(a_i = 4) + \beta_6 e_i + \beta_7 g_i + \beta_8 l_i, \sigma).$$

Still, $\text{normal}(0, 1)$ is assigned as prior to all coefficients and $\text{normal}^+(0, 1)$ is assigned to $\sigma$ as prior.

### D.6. `radon`

The dataset contains data of radon levels in 386 different counties in the USA (Gelman, 2007). $n = 12573$ measurements are collected. In the $i$th measurement, the county id $c_i \in \{1, \ldots, 386\}$, the floor measure $f_i \in \{0, \ldots, 9\}$ and the log-radon measurement $r_i$ are collected. We work with the model

$$r_i \sim \text{normal}(\alpha_{c_i} + \beta_{c_i} f_i, \sigma).$$

Each $\alpha_\cdot$ is assigned the prior $\text{normal}(\mu_\alpha, \sigma_\alpha)$ and each $\beta_\cdot$ is assigned the prior $\text{normal}(\mu_\beta, \sigma_\beta)$. Furthermore, $\mu_\alpha$ and $\mu_\beta$ are both assigned the prior $\text{normal}(0, 10)$. $\sigma, \sigma_\alpha, \sigma_\beta$ are all assigned the prior $\log\_\text{normal}(0, 1)$.

### D.7. `wells`

The dataset studies factors affecting the decision of households in Bangladesh to switch wells (Gelman, 2007). There are in total $n = 3020$ datapoints. Each datapoint $i$ includes the arsenic level $a_i \in \mathbb{R}^+$, the distance to the closest known safe well

$d_i \in \mathbb{R}^+$, years of education of household head $e_i \in \mathbb{R}^+$ and decision of switching the well $s_i \in \{0, 1\}$. The regression model is

$$s_i \sim \text{Bernoulli}(\text{Logit}^{-1}(\beta_0 + \beta_1 a_i + \beta_2 d_i + \beta_3 e_i + \beta_4 a_i d_i + \beta_5 a_i e_i + \beta_6 d_i e_i)).$$

After centering the data, the coefficients are all assigned $\text{normal}(0, 1)$ as priors.

