# OpenReview forum: "Predictive variational inference: Learn the predictively optimal posterior distribution"
_ICML.cc/2026/Conference — ICML 2026 regular_

### Official Review · Reviewer_s3Kn · 2026-03-09

**Soundness:** 2
**Presentation:** 3
**Significance:** 2
**Originality:** 3
**Overall Recommendation:** 4
**Confidence:** 3

**Summary:**

The paper proposes Predictive Variational Inference (PVI), a framework for performing variational inference under a predictive-oriented objective rather than approximating the Bayesian posterior. Instead of minimizing a divergence between $q(\theta)$ and the posterior $p(\theta \mid y)$, the method optimizes a scoring rule applied to the posterior predictive distribution, aiming to produce a predictive distribution closer to the true data-generating process under potential model misspecification. The authors develop gradient estimators for several scoring rules and demonstrate the approach on several examples including logistic regression, election modeling, and a likelihood-free cryoEM example.

**Compliance With Llm Reviewing Policy:**

Affirmed.

**Final Justification:**

I appreciate the authors’ feedback, especially their clarification regarding the uniqueness issue and the dimensionality of the experiments. While I remain not fully convinced by the discussion on optimization, I recognize that this reflects a broader theoretical limitation in the current literature rather than a paper-specific weakness. Overall, the rebuttal has addressed part of my concerns, and I have therefore adjusted my score accordingly.

**Key Questions For Authors:**

See Weakness.

**Strengths And Weaknesses:**

The paper is very well-written, and its definition of notations are smooth and clear.

# Unclear properties of the predictive optimum $\phi^*$
The theoretical development introduces the predictive optimal parameter $\phi$, but it is unclear whether this solution is unique. In general, the predictive objective (Eq. 3) may admit multiple solutions, especially when the variational family is flexible. Some regularity assumptions needed for proposition 1 appear to implicitly require uniqueness, but these conditions are not clearly stated or discussed. Do these conditions depend on the uniqueness of $\phi^*$?

# Empirical evidence is weak
A weakness of the empirical evaluation is that most experiments are conducted in very low-dimensional parameter settings. For instance, the toy normal example and the cryoEM application both involve essentially one-dimensional latent parameters, and the golf putting model contains only a small number of regression coefficients. While these examples illustrate the conceptual behavior of PVI, they do not demonstrate how the method scales to high-dimensional inference problems, which are the primary motivation for variational inference methods. It would strengthen the paper to include experiments with moderately or high-dimensional latent variables (e.g., neural network models or high-dimensional latent variable models) to assess the practical scalability and robustness of the proposed approach.

# Weak optimization analysis
The optimization aspect of the paper is relatively weak, although it comprises the main contribution of this paper. While gradient estimators are derived for different scoring rules, the paper does not provide a detailed analysis of the convergence of the optimization procedure. A deeper analysis of its optimization properties and convergence behavior would strengthen the technical contribution of this paper.

# More intricate analysis of optimization as Wasserstein gradient flow
The paper focuses on optimizing the objective with respect to a parametric variational family $q_\phi$. However, the objective in Eq. (3) is naturally defined over the space of probability measures. It would be interesting to consider whether the optimization could instead be performed nonparametrically, for example via a Wasserstein gradient flow directly over distributions $q$. Such a formulation could potentially lead to stronger theoretical guarantees on convergence. Another advantage is that it would not require a unique parameter $\phi^*$ but only a unique distribution $q^\ast$.

---

> ### Author Rebuttal · Authors · 2026-03-31
>
> We are grateful for your comments. We hope our reply addresses your concerns.
> ## Unique optimum
> First, although our current Prop.1 is stated with a unique optimizer for clarity,  it essentially uses the classic M-estimation theory, and the asymptotic consistency **does not rely on the uniqueness** of the optimizer. Define $$ Q^\*=\\arg\\max\_{q\\in\\mathcal{Q}}S(\\int p(\\cdot|\\theta)q(\\theta)d\\theta,p\_{true}(\\cdot)),$$ the set of predictive-optimal variational distributions. The consistency can be stated as a weak convergence of the induced distributions $q\_{\\phi}$ to the set $Q^\*$.
>
> Second, although the consistency holds without uniqueness, it may be of independent interest to understand when uniqueness holds. We distinguish three levels: the uniqueness of (a) variational parameter, (b) variational distribution, and (c) optimal predictive distribution. Under mild conditions (e.g., $y$ continuous on $R^d$ and $p(y|\\theta)$ continuous and strictly positive), (c) is unique. In contrast, (a) typically fails for flexible flows since flow itself is non-identifiable. Conditions for (b) are less clear and relate to integral equation theory.
>
> Third, our proposed prior/posterior regularization partially addresses the potential non-uniqueness at (b). The above discussion considers only the predictive objective. The regularization term has a unique optimum, and with an appropriate level of regularization, it serves as a constraint on the optimization problem to break ties.
> ## Extended empirical evidence and model complexity
> Some “demonstration” examples (golf, cryoEM) are “easy” in dimension counts (despite this, cryoEM is nontrivial due to its intractable likelihood). This design was intentional, allowing us to illustrate the salient features of the method.
>
> We apologize if this was not sufficiently clear: The paper also includes extended benchmarking experiments (Supp. Tab 2\) on a complex survey (Sec 4.2), and a collection of regression models in the PosteriorDB database, in which “radon” contains 777 parameters; “glmm” contains 247 parameters and 2,072 observations, demonstrating the practical feasibility in applied Bayes.
>
> Let us further clarify on model complexity. First, the motivation of VI is distinct from its usual use as a tool for approximating large models. Our primary objective of interest is the predictive posterior and associated uncertainties, with the computational tools serving to implement the methodology in a scalable way.
>
> Second, while classical ML approaches increase “model” complexity with fixed “inference”, we explore the orthogonal direction: to make “inference” deeper. Even when $\\theta$ is one-d, to infer the optimal $q(\\theta|y)$ is infinite-dimensional.
>
> Last, as in Sec 2.4, PVI is an implicit hierarchical model. As PVI effectively marginalizes out local parameters, it often avoids costly latent variable formulations. For instance, the original cryoEM model $p(y\_i|\\theta)$ assumes a fixed angle $\\theta$ across all images. A fully hierarchical model $p(y\_i|\\theta\_i)$ requires per-image inference $\\theta\_i$, infeasible at scale. PVI retains the same global parameter $\\theta$, while achieving the same inferential goal: the population distribution of $\\theta\_i$.
> ## Optimization analysis
> Part of our contribution is to derive unbiased gradient estimates given our new PVI objectives (e.g., a novel unbiased gradient in the log score). We prove they are unbiased, and the SGD convergence follows standard results. However, a mathematical analysis of the expected training dynamics is likely beyond our scope, as understanding the behavior of flow training is difficult even for toy problems.
>
> Empirically, we generally did not run into serious numerical issues. Since our framework allows flexible choices of scoring rules, it is now possible to compare scoring rules via the lens of optimization stability in Bayesian learning, which we leave for a future direction.
> ## Concurrent work
> We apologize for not thoroughly discussing concurrent work on Wasserstein gradient flows.  Our approach is complementary: we focus on tractable parametric variational implementations.
>
> While gradient flows can fit some simpler cases, they can be harder to scale for some of our benchmark datasets. As McLatchie et al. noted,
>
> > Naive implementations are infeasible for high-dimensional problems, generally computationally demanding, and require careful tuning of several hyperparameters.
>
> Our PVI supports a spectrum of approximations: The full normalizing-flow PVI handles moderate-dimensional regressions well, while simpler families (e.g., mean-field Gaussian) further improve scalability in higher-d.
>
> Moreover, our framework provides a general recipe applicable to a wide range of proper scoring rules, which is less straightforward in the gradient flow formulations.
>
> We view these two concurrent developments as complementary. A formal and deeper comparison will be promising, but beyond the present scope.

---

> > ### Author Rebuttal · Reviewer_s3Kn · 2026-03-31
> >
> > I appreciate the authors’ feedback, particularly their clarification regarding uniqueness and the dimensionality of the experiments. I am still not fully convinced by the discussion around optimization, but I recognize that this concerns a broader theoretical gap in the current literature rather than a flaw specific to this paper. In light of the rebuttal, I have adjusted my score accordingly.

---

### Official Review · Reviewer_REdi · 2026-03-10

**Soundness:** 3
**Presentation:** 3
**Significance:** 3
**Originality:** 4
**Overall Recommendation:** 5
**Confidence:** 4

**Summary:**

This paper suggests a "predictive" version of variational inference, where the variational objective is based on a scoring rule measuring the discrepancy between the variational posterior predictive and the data generating process, in combination with a regularization term either to the prior or the posterior. As a result, the obtained variational posterior performs quite differently to Bayes, as it may not collapse to a point under model misspecification.

**Compliance With Llm Reviewing Policy:**

Affirmed.

**Final Justification:**

The rebuttal has addressed my concerns regarding the theory and intuition for the method, and I have increased my score appropriately. This is an interesting paper that offers a new perspective on variational inference with a focus on prediction.

**Key Questions For Authors:**

1. The conditions for Proposition 1 are a quite unrealistic, which the authors highlight themselves in Appendix A.1, especially around the uniqueness of $\phi_0$ which seems to not hold for most interesting examples. I think it is quite important that this comment and counterexample be moved to the main paper, as the theoretical result is a bit misleading otherwise. For more interesting examples, I doubt we will have asymptotic normality, but is consistency still expected for say a local maximum? In any case, I think Proposition 1 should really be surrounded by a warning, and I also think claims in the paper surrounding the theory (e.g. in the discussion) might want to be toned down a bit as a result.

2. I think the connection to hierarchical Bayes/model expansion is quite interesting, but I don't quite follow what is meant by an "implicit-hierarchical" view. Could the authors please expand upon what they meant by this?

3. Does regularization to the prior or the posterior work better in practice? It is a bit unclear what is actually implemented in practice in the experiments. Does regularizing to the posterior also cause an issue of "double dipping" from the data?

4. There seems to have been more recent work in this area, which was not discussed in the related work sections, such as Shen et al. (2025) and McLatchie et al. (2025), the latter of which is only given a brief mention at the end. I acknowledge that the timeline of preprints make this a bit challenging, but it would still be good for the reader for these works to at least be mentioned in related works as they were available at the time of submission of this version of the paper.

Shen, Z., et al. "Prediction-Centric Uncertainty Quantification via MMD." Proceedings of The 28th International Conference on Artificial Intelligence and Statistics. 2025.

**Limitations:**

Yes.

**Strengths And Weaknesses:**

Strengths: The paper is well-written, and there are some novel and interesting ideas here surrounding an alternative role of the variational posterior as a mixing distribution for the predictive. Ideas related to this paper, and recent work building off it, has picked up some interest in the Bayesian literature recently as well.

Weaknesses: The theory is on the weak side, as the assumptions for the main result are quite unrealistic (which the authors highlight themselves). There is also some related work that seems to be missing from the related work section. A few areas of the writing could also be made more clear.

---

> ### Author Rebuttal · Authors · 2026-03-31
>
> ## Consistency and uniqueness
>
> Thank you for this important comment. We will move Appendix A.1 discussions to the main text to clarify.
>
> 1. When consistency holds. Although current Prop.1 is stated with a unique optimizer for easy reading, it essentially uses the classic M-estimation theory, and the asymptotic consistency **does not rely on uniqueness** of the optimizer.  Define $$Q^\*=\\arg\\max\_{q\\in\\mathcal{Q}}S(\\int p(\\cdot|\\theta)q(\\theta)d\\theta,p\_{true}(\\cdot)),$$ the set of predictive-optimal variational distributions. The consistency is stated as a weak convergence of the induced distributions $q\_{\\phi}$ to the set $Q^\*$. However, it is a limitation that asymptotic normality typically fails without local identification. We apologize for not having this clarification.
> 2. When uniqueness holds. Although consistency holds without uniqueness, it may be of independent interest to know when the uniqueness holds. We distinguish three levels: the uniqueness of the optimal (a) variational parameter, (b) variational distribution, and (c) induced predictive distribution. Under mild conditions (e.g., $y$ continuous on $R^d$ and $p(y|\\theta)$ continuous and strictly positive), (c) is unique. In contrast, (a) typically fails for flexible flows since flow itself is non-identifiable. Conditions for (b) are less clear and mathematically relate to integral equation theory.
> 3. The regularization matters. The above discussion considers only the predictive objective. Our proposed prior/posterior regularization is partially designed to address the potential non-uniqueness at (b). The regularization term has a unique optimum, and with an appropriate level of regularization, it serves as a constraint on the optimization problem to break ties.
>
>
> ## Implicit hierarchical
>
> We thank you for the observation on the connection to hierarchical Bayes, which we view as a conceptual contribution.
>
> Take our cryoEM example for illustration. The original model $p(y\_i|\\theta)$, used in scientific literature, assumes a fixed binding angle $\\theta$ across all images (complete pooling).  A hierarchical model $p(y\_i|\\theta\_i)$ infers per-image angles $\\theta\_i$. In contrast, PVI retains the same global parameter $\\theta$, while it achieves the same inferential goal: the population distribution of $\\theta\_i$.
>
> In general, heterogeneity is a reason for model misspecification. To correct for it, hierarchical Bayes lets parameters vary. There are proposals (e.g., Wang and Blei 2018\) to hierarchicalize any model, explicitly expanding $\\theta$ to $\\theta\_i$. We call PVI “implicit hierarchical”: while allowing parameters to vary in the population, it effectively marginalizes local parameters and avoids the full hierarchical cost.
>
> Further in Sec 4.2, the uncertainty in $q(\\theta)$ indicates where extra hierarchical structure is needed, so that PVI guides hierarchical model building.
>
> ## Regularization
>
> In benchmark experiments, we run cross-validation to decide between the prior and posterior regularization. Across PosteriorDB regression models, we did not find a clear pattern for whether the prior or posterior regularization works better (Tab. 3, [anonymized link](https://anonymous.4open.science/r/pvi_icml-064D/reb.pdf)). We view this extra binary tuning as added flexibility.
>
> Both prior and posterior regularizations are justified: the prior regularization term appears in regular VI and other generalized Bayes. The posterior regularization interpolates between the full Bayes and full predictive. We are not worried about “double-dipping”. PVI contrasts with multiplying two inferences (which adds information and shrinks uncertainty). In PVI, the posterior is a shrinkage baseline, and the predictive objective remains the primary driver, allowing the method to infer the correct uncertainty. This is an advantage of our work and we will elaborate in the revision.
>
> ## Concurrent work
>
> We apologize for not thoroughly referencing concurrent works on Wasserstein gradient flows. For completeness, the results presented here were arXiv’d prior to McLatchie et al.
>
> Further, our work is complementary: we focus on tractable parametric variational implementations, not gradient flows.
>
> Shen et al. focus on MMD, while our framework provides a general recipe for a wide range of scoring rules, which is less straightforward in the gradient flow formulation.
>
> While gradient flows fit simpler cases, they can be harder to scale for some of our benchmark models. As McLatchie et al. noted,
>
>  > Naive implementations are infeasible for high-dimensional problems, generally computationally demanding, and require careful tuning of several hyperparameters.
>
> Our PVI permits a spectrum of approximations: The normalizing-flow PVI handles moderate-dimensional regressions well, while simpler families (e.g., mean-field Gaussian) further improve scalability in higher-d.

---

> > ### Author Rebuttal · Reviewer_REdi · 2026-04-01
> >
> > Thank you for addressing my questions. I still have some remaining doubts about consistency in the case where there is no unique optima, as I suspect some additional regularity conditions might be required given the general non-identifiability of mixtures. Nonetheless, I acknowledge this is beyond the scope of the paper. I have raised my score accordingly.

---

### Official Review · Reviewer_qD43 · 2026-03-11

**Soundness:** 4
**Presentation:** 3
**Significance:** 3
**Originality:** 3
**Overall Recommendation:** 5
**Confidence:** 4

**Summary:**

This paper proposes, predictive variational inference (PVI), in which a variational approximation is optimized for predictive accuracy rather than parameter recovery. This leads to a procedure with very different statistical properties, such as not concentrating when the model is misspecified. The paper proposes a number of different scoring rules that can be used to construct the PVI loss function along with accompanying gradient estimators and supporting theory. The authors show that with the correct choice of loss function, PVI can be used for simulation-based inference and also being robust to model misspecification. They also provide an interpretation of PVI as a kind of nonparametric hierarchical model. Experiments demonstrate how PVI can be used to detect model misspecification and diagnose parts of the model that require elaboration.

**Compliance With Llm Reviewing Policy:**

Affirmed.

**Final Justification:**

The authors addressed my concerns and I have raised my score

**Key Questions For Authors:**

1. Which scoring rules were used in the experiments?
2. How do the different scoring rules compare in terms of performance? Do they lead to similar or very different PVI posteriors when applied to the same problem?
3. How good is the predictive performance of PVI compared to alternatives?
4. How does the computational cost of PVI compare to alternatives? How does it depend on the choice of scoring rule?

Satisfactory answers to these questions would lead me to raise my soundness, presentation, and significance scores, and hence improve my overall recommendation.

**Limitations:**

yes

**Strengths And Weaknesses:**

### Strengths And Weaknesses

**Soundness:** The paper provides a good amount of supporting theory and a nice variety of well thought-out experiments. However, there are a few holes in the evaluations. First, while four scoring rules are suggested, they are never compared. Second, there are no results concerning the predictive performance of PVI. Third, there is no discussion of computational cost, and how it compares to vanilla VI or MCMC.

**Presentation:** The presentation is mostly clear. However, there are many typos and grammatical errors that did make reading the paper more challenging. There were also some inconsistencies such as in the use of IID and i.i.d. Finally, Table 1 hard to read (please add lines or more separation between the rows).

**Significance:** PVI has tremendous potential as a method. However, the significance of this paper is limited by the missing empirical comparisons discussed under "Soundness."

**Originality:** While similar to concurrent work of McLatchie et al. (2025), the paper is nevertheless quite distinct, in it uses different scoring rules, a different approach to approximate inference, focuses on implications for parameter inference / diagnosis of mispecification, and introduces an insight hierarchal modeling interpretation.

---

> ### Author Rebuttal · Authors · 2026-03-31
>
> We greatly thank the reviewer for the suggestions. We add new experiments as suggested. We hope our responses and extra empirical results can clarify these points raised.
>
> ## Which scoring rules
>
> We apologize for not making the experimental details (currently in Supplement) clear in the main text. We will add details in the revision. We used CRPS in the cryoEM example, which is naturally compatible with likelihood-free inference. In the other examples in the main paper we use the log score. In Supplement, we run PVI under all four scores in Table 1 (log, quadratic, CRPS, IS) on PosteriorDB benchmarks.  We view this generality as an important benefit.
>
> ## Compare scoring rules
>
> In Supp. Table 2, we have looked at how PVI/VI perform under different scoring rules, partially to see whether PVI improves on classic inference and partially to see whether training on one of the scoring rules can outperform evaluation with other scoring rules. A scoring rule is often a user choice (e.g., to care about calibrated uncertainties then CRPS or IS are natural).  We find generally that if a user wants to use a particular scoring rule, then they are best served by using PVI with the associated scoring rule. See also Supp. B.3 and B.5.
>
> Because PVI can incorporate a flexible choice of scoring rules, a fruitful direction for future work would be to compare scoring rules via the lens of inference efficiency, robustness, and optimization stability. To our knowledge, such comparisons were not systematically studied, and perhaps one value of PVI is to provide a unified framework for them.
>
> We empirically observe that the posteriors from different scoring rules usually overlap with each other but are not exactly the same. At a high level, this solution diversity reflects the benefit of multiple scoring rules. To be concrete, we have added new results (Figure 2 in the [anonymized link](https://anonymous.4open.science/r/pvi_icml-064D/reb.pdf)), comparing PVI solutions in the “earnings” model under various scoring rules.
> While most posteriors have similar means (e.g., $\\beta\_0$  is close to 9), the posterior uncertainty differs across scoring rules, each serving a different need in prediction.
>
> ## Compare PVI to alternatives
>
> The premise of PVI is improved predictive performance. We have verified this advantage using the benchmark models in PosteriorDB. Table 2 in the Supplement shows the expected behavior relative to the standard alternative, vanilla VI using the same variational approximation. As expected, when evaluated using a proper scoring rule on real datasets, the PVI posterior adapted to that scoring rule produces the best performance (with the single exception that vanilla VI happens to perform very well under IS on the “radon” dataset, indicating the model is already close to well-specification; it is consistent with our view of further using PVI for a model checking tool).
>
> For completeness, we also add results using Hamiltonian Monte Carlo (NUTS) with the exact posterior as another alternative. This additional experiment is shown as Table 1 of the [anonymized link](https://anonymous.4open.science/r/pvi_icml-064D/reb.pdf). We do not see a qualitative jump in performance from VI to NUTS.
>
> ## Computation cost
>
> First, PVI conceptually has a higher computation cost than vanilla VI, as it shifts inference from a finite-dimensional parameter $\\theta$ to the infinite-dimensional space of distributions $q(\\theta)$. Although we agree that this added cost may limit scalability to very high-dimensional problems (e.g., Bayesian neural nets),  we find it generally feasible in applied Bayesian data analysis, such as the PosteriorDB benchmarks. Additionally, PVI permits a spectrum of approximations: simpler families (e.g., mean-field Gaussian) further improve scalability in higher dimensions.
>
> We also clarify that PVI is not intended as a cheaper approximation to Bayes; rather, it provides a more expressive solution for improved predictive performance. Unlike classical ML approaches that increase “model”’ complexity with fixed “‘inference”, we use PVI to explore the orthogonal direction: making “inference” deeper.
>
> *Cost of different score rules*: In terms of per iteration time, PVI with log score is comparable to vanilla VI. However, PVI with CRPS or IS is slower. To illustrate this empirically, we include additional results on the “earnings” model, showing convergence behavior and training time under the log score, CRPS, and IS (Fig. 1 and Table 2 in the  [anonymized link](https://anonymous.4open.science/r/pvi_icml-064D/reb.pdf)).
>
> Additionally, to your comments in “Presentation”. We will clean up the table and fix any inconsistencies. For the “lack of comparisons” concerns raised under “Soundness”,  we apologize for not presenting them more clearly, and hope the explanations above help clarify. We particularly thank you for distinguishing our work from concurrent ones.

---

> > ### Author Rebuttal · Reviewer_qD43 · 2026-03-31
> >
> > Thanks for the authors for their thorough response, and the many additional results in the Appendices. I have increased my score.

---

### Official Review · Reviewer_Lkbk · 2026-03-13

**Soundness:** 4
**Presentation:** 3
**Significance:** 4
**Originality:** 3
**Overall Recommendation:** 5
**Confidence:** 3

**Summary:**

They propose optimizing a kind of prequential loss that matches a variational posterior in a one-step-ahead fashion instead of to the true posterior.  This introduces various difficulties with unbiased gradients that are addressed under some circumstances.  They demonstrate on a few different types of problem, and draw connections to related methods such as PAC-Bayes

**Compliance With Llm Reviewing Policy:**

Affirmed.

**Key Questions For Authors:**

1) "we keep λ as a small fixed constant" this seems suspicious - how did you actually choose it?
2) "We show that this novel gradient estimate (9) is always unbiased, regardless of the batch size" - even if no sample is accepted and you redraw until you get an acceptance?
3) Can you talk about to what extent "Predictively Oriented Posteriors" is concurrent work?
4) It's great that this is a gradient-based method, but it was never quite clear to me that there isn't a gotcha in scaling.  What was the largest number of variational parameters fit with this method?

**Limitations:**

yes

**Strengths And Weaknesses:**

Strengths:
==============
1) This is a very central problem in model fitting in general!
2) There was a nice variety of demos and motivating problems,
3) I liked the extensive connections made to other methods such as PAC-Bayes, and the breadth of the literature drawn upon was large.
4) There are lots of supplementary experiments.


Weaknesses:
==============
1) There was no nice algorithm box showing how the updates compare to standard VI on some simple examples.  I also didn't understand what limitations are placed on using a optimized
2) It's not totally clear how limiting all the constraints are on this method when put together.

---

> ### Author Rebuttal · Authors · 2026-03-31
>
> We thank the reviewer for the positive assessment and helpful feedback. We will follow your suggestions in the revision. Below is our response to your comments.
>
> ## Added algorithm box
>
> We agree an algorithm box would improve clarity and will include one in the revision.
>
> We also emphasize that our contribution is a unified framework covering *a family of* algorithms across different scoring rules, with gradient estimators derived for each. We view this generality as a key strength of the work.
>
> ## The regularization strength $\\lambda$
>
> We apologize for not making the experimental detail clear in the main text and we will move key details from the appendix to improve clarity.
>
> In illustrative **toy examples** we used in the manuscript, we did not apply regularization, as our goal was to illustrate the methodology on a couple of relatively transparent examples. That is, regularization was not employed to improve predictive performance. This is our intentional design so that we do not hide the intrinsic behavior of the method through premature optimization.
>
> In more realistic examples and our PosteriorDB benchmarking datasets (Supplement), we treated the choice of $\\lambda$ and choice of whether to regularize towards the prior or posterior as tuning parameters, and we use cross-validation on a grid of $\\lambda \\in \\{0, 0.01, 0.1, 1.0\\}$. We provided tuning details in Supplement. Like most regularization methods, we recommend mild tuning in practice.
>
> ## Unbiasedness of the gradient estimator
>
> Yes, the gradient estimator based on rejection sampling until acceptance is unbiased. If no sample is accepted, we redraw until acceptance.
>
> Formally, this can be viewed as sampling at a stopping time (first acceptance), and the resulting draw has the correct conditional distribution. Unbiasedness then follows from the law of total expectation.
>
> ## Concurrent work on Wasserstein gradient flows
>
> McLatchie et al. (2025) applies the same basic idea as this paper, deriving a pseudo-posterior for different scoring rules, but does so without using a variational characterization and instead uses Wasserstein gradient flow to approximate the exact optimal choice of predictive distribution; they also engage more directly with the theoretical properties of the model, establishing a contraction rate in the well-specified setting. We think it is a very good paper.
>
> We view these two works as concurrent. For completeness, the results presented here were on arXiv prior to McLatchie et al.
>
> Furthermore, our approach is complementary in implementations: we focus on tractable parametric variational implementations, not gradient flows. Moreover, our framework provides a general recipe applicable to a wide range of proper scoring rules, which is less straightforward in the gradient flow formulations.
>
> ## Scalability and parameter size
>
> The number of variational parameters we use could be quite large, as our choice of variational family includes a normalizing flow; those experiments (e.g., the one in Appendix B.2) used a neural spline flow with 8 knots, 32 hidden units, 3 hidden layers and 4 flow blocks.
>
> A related question is whether there is any upper limit on the dimension of $\\theta$. The largest number of parameters in our PosteriorDB benchmark examples is the “Radon” model in Table 2, which is 777, demonstrating the practical feasibility in applied Bayesian data analysis.
>
> We note as well that, in view of our interpretation of PVI as an implicit hierarchical expansion, even in the case where we use a small number of parameters we are effectively running an infinite dimensional inference $q(\\theta)$ in the space of probabilistic distributions. While classical ML approaches increase “model” complexity with fixed “‘inference”, we explore the orthogonal direction: to make “inference” deeper.

---

> > ### Author Rebuttal · Reviewer_Lkbk · 2026-04-03
> >
> > This answers all my questions, except I still am not totally confident you could use this method to fit an approximate posterior with millions of parameters.  (seems likely, but I would have liked a demo)

---

> > > ### Author Response · Authors · 2026-04-06
> > >
> > > We thank the reviewer for noting that our previous responses were helpful. As requested, we have added a new experiment in higher dimensions. We hope this new empirical demonstration and our answers below help address the follow-up question on scalability.
> > >
> > > First, we agree that demonstrating scalability to very large models (e.g., neural nets with millions of parameters) would further strengthen the work. At the same time, our focus is on practical Bayesian data analysis, where typical benchmark problems (e.g., PosteriorDB models) have dimensionality on the order of <10^3. Within this regime, our paper demonstrates that PVI is feasible.
> > >
> > > More broadly, PVI inherits the scalability properties of the underlying variational family. In particular, using simpler families (e.g., mean-field Gaussian), it can scale to higher dimensions.  As a check, we implemented log-score PVI in a normal model with parameter dimension up to 10,000 (see Fig. 1 in the [anonymized link](https://anonymous.4open.science/r/pvi_icml-064D/res.pdf)). We considered a model $p(y \mid \theta) = \mathrm{MVN}(\theta, I_d)$ with IID data $y_i \sim \mathrm{MVN}(0, 2 \\; I_d)$ and used a mean-field Gaussian variational family $(n=100, M=10)$. As the dimension of $\theta$ increases, the expected log predictive density (elpd; higher is better) scales linearly for both VI and PVI. Across all dimensions, PVI consistently achieves higher elpd than VI, indicating improved predictive performance. While this experiment is intentionally simple due the time limit of the rebuttal, it demonstrates that PVI can be applied in higher-dimensional settings. Of course, it remains an important direction for future work to practically pushing scalability of PVI in modern deep models.
> > >
> > > Further, fitting full Bayesian models with millions of parameters remains challenging even for standard approaches (e.g., MCMC or VI). In particular, fitting expressive variational families such as normalizing flows at this scale is itself an open problem.
> > >
> > > Finally, we emphasize that PVI shifts complexity from the model to the inference. The role of variational inference here differs from its conventional use for approximating large models: our primary objective is the predictive posterior and its associated uncertainties. As discussed in Sec. 2.4, PVI can be viewed as an implicit hierarchical model; by effectively marginalizing local parameters, it can avoid costly latent variable formulations and large parameter spaces.

---

### Decision · Program_Chairs · 2026-04-30

**Decision:**

Accept (regular)

**Comment:**

The paper proposes a Predictive Variational Bayes framework that directly optimises a posterior for predictive performance under proper scoring rules. It is shown to be quite relevant as it is known that under model misspecification, exact Bayes or standard variational approximations may not be optimal.

The reviewers find the paper well written and sound. They also agree that the paper is potentially impactful. Further, the rebuttal addresses almost all critical comments satisfactorily. The only unresolved comment concerns an ultra-high-dimensional application to models such as Variational neural networks. Such an application is expected in general for ICML, and it would be great to add it to the camera-ready version. Although not needed per se, as the reviewer was happy with the rebuttal.

In my opinion, the paper is highly relevant and original. I appreciated a set of used examples indeed. That said, for the camera-ready version, I would like to ask the authors to make positioning wrt Generalised and predictive Bayes already in the Introduction. I would also ask to clean up minor notation issues and be consistent with distribution notation: perhaps stick to N(,) for univariate normal and MVN(,) for multivariate normal. Perhaps add links to the usefulness of PVI in the context of multilevel regression with post-stratification applications, where within-strata cell-level predictive distributions are crucial.  Perhaps recommendations on using PVI within the Bayesian workflow could be made in one of the appendices, too.

Overall, I support an accept decision.